# A probabilistic Taylor expansion with Gaussian processes

**Toni Karvonen**                                        *toni.karvonen@helsinki.fi*
*Department of Mathematics and Statistics*
*University of Helsinki*

**Jon Cockayne**                                        *jon.cockayne@soton.ac.uk*
*Department of Mathematical Sciences*
*University of Southampton*

**Filip Tronarp**                                        *filip.tronarp@matstat.lu.se*
*Centre for Mathematical Sciences*
*Lund University*

**Simo Särkkä**                                        *simo.sarkka@aalto.fi*
*Department of Electrical Engineering and Automation*
*Aalto University*

**Reviewed on OpenReview:** *https: // openreview. net/ forum? id= 2TneniEIDB*

## Abstract

We study a class of Gaussian processes for which the posterior mean, for a particular choice of data, replicates a truncated Taylor expansion of any order. The data consist of derivative evaluations at the expansion point and the prior covariance kernel belongs to the class of Taylor kernels, which can be written in a certain power series form. We discuss and prove some results on maximum likelihood estimation of parameters of Taylor kernels. The proposed framework is a special case of Gaussian process regression based on data that is orthogonal in the reproducing kernel Hilbert space of the covariance kernel.

## 1 Introduction

Taylor's theorem is among the most fundamental results in analysis. In one dimension, Taylor's theorem states that any function $f \colon \mathbb{R} \to \mathbb{R}$ that is sufficiently smooth at $a \in \mathbb{R}$ can be written as

$$f(x) = T_{n,a}(x) + P_{n,a}(x), \tag{1.1}$$

where $T_{n,a}(x) = \sum_{p=0}^{n} \frac{1}{p!} f^{(p)}(a)(x-a)^p$ is the $n$th order Taylor polynomial and $P_{n,a}(x)$ a remainder term which has the property that $P_{n,a}(x) = \mathcal{O}(|x-a|^{n+1})$ as $|x-a| \to 0$. Multidimensional generalisations are readily available and will be introduced in Section 2. Approximations derived from (1.1), in particular the first and second order Taylor approximations

$$f(x) \approx f(a) + f'(a)(x-a) \quad \text{and} \quad f(x) \approx f(a) + f'(a)(x-a) + \frac{1}{2}f''(a)(x-a)^2,$$

play an important role in numerical algorithms for a number of reasons. Firstly, Taylor approximations provide a straightforward and principled means of *linearising* a function of interest, which can often dramatically accelerate otherwise costly computations. Secondly, they require only information about a function and its derivatives at a single point; information that particular algorithms may already be collecting. Particular applications of Taylor's theorem in numerical algorithms include optimisation (Moré, 1978; Conn et al., 2000), state estimation (Särkkä, 2013, Ch. 5), ordinary differential equations (Hairer et al., 1993, Ch. II), and approximation of exponential integrals in Bayesian statistics (Raudenbush et al., 2000), to name but a few.

A crucial challenge when applying Taylor series in this way, however, is their locality. The approximation is valid only near $a$, and apart from trivial examples approximation quality decays rapidly away from this point. When a numerical algorithm attempts to use a Taylor approximation to explore function behaviour around a particularly novel point, far from $a$, the behaviour of the algorithm can be difficult to predict and control.

This paper proposes a remedy for this by introducing a Gaussian process (GP) model (Rasmussen and Williams, 2006) whose posterior mean, given the *derivative data* $(f(a), f'(a), \ldots, f^{(n)}(a))$, is exactly the Taylor polynomial $T_{n,a}$, and whose posterior variance plays a role analogous to the remainder term $P_{n,a}$. In the spirit of probabilistic numerics (Diaconis, 1988; Cockayne et al., 2019b; Hennig et al., 2022), the posterior variance can then be used for principled probabilistic quantification of epistemic uncertainty in the Taylor approximation $f(x) \approx T_{n,a}(x)$ at $x \neq a$, which can be exploited and propagated forward. In effect, the variance may be used to encode into algorithms a degree of "scepticism" about the validity of the Taylor approximation away from $a$. Taylor approximation thus joins the ranks of classical numerical methods, such as algorithms for spline interpolation (Diaconis, 1988; Kimeldorf and Wahba, 1970), numerical quadrature (Diaconis, 1988; Karvonen and Särkkä, 2017; Karvonen et al., 2018), differential equations (Schober et al., 2014; 2019; Teymur et al., 2016), and linear algebra (Cockayne et al., 2019a; Hennig, 2015), that can be cast as statistical inference. Even though the use of derivative information in Gaussian process modelling is rather standard and the prior that we use is relatively well known, we are unaware of any prior attempts at deriving the Taylor approximation in a Gaussian process framework.

## 1.1 Related Literature

The Gaussian process priors we use to construct a probabilistic Taylor expansion are determined by positive-definite and non-stationary *Taylor kernels* which, at an expansion point $a$, take the form

$$K_a(x,y) = K(x-a, y-a), \quad \text{where} \quad K(x,y) = \sigma^2 \sum_{p=0}^{\infty} \frac{c_p \lambda^p}{(p!)^2}(xy)^p \tag{1.2}$$

for non-negative constants $c_p$ and positive parameters $\sigma$ and $\lambda$. For multidimensional input points, the index $p \in \mathbb{N}_0$ is replaced with a multi-index $\boldsymbol{\alpha} \in \mathbb{N}_0^d$ and $xy$ usually with the Euclidean inner product; see Section 2.1 for details. The canonical example is the *exponential kernel*

$$K(x,y) = \sigma^2 \sum_{p=0}^{\infty} \frac{\lambda^p}{p!}(xy)^p = \sigma^2 \exp(\lambda xy), \tag{1.3}$$

which is obtained by setting $c_p = p!$ in (1.2). The exponential kernel is closely connected to the popular Gaussian kernel $K(x,y) = \sigma^2 \exp(-\lambda^2(x-y)^2/2)$.

Taylor kernels, often under the name *power series kernels*, have been used—and their approximation properties analysed—in the numerical analysis and scattered data approximation literature; see Dick (2006); Zwicknagl (2009); De Marchi and Schaback (2010); Zwicknagl and Schaback (2013) and Fasshauer and McCourt (2015, Sec. 3.3.1). Section 1 in Zwicknagl and Schaback (2013) has been of particular inspiration for the present work. The *Szegő kernel* and the *Bergman kernel* $K(x,y) = 1/(1-xy)$ and $K(x,y) = 1/(1-xy)^2$, respectively, are particularly well studied in the approximation theory literature because their reproducing kernel Hilbert spaces (RKHSs) are important in complex analysis; see, for example, Larkin (1970); Richter-Dyn (1971b;a) and Oettershagen (2017, Sec. 6.2). These kernels are defined on the open interval $(-1, 1)$ and obtained from (1.2) by setting (Szegő) $c_p = (p!)^2$ and (Bergman) $c_{2p} = (p!)^2$ and $c_{2p+1} = 0$ for $p \in \mathbb{N}_0$ in (1.2).

Taylor kernels occasionally appear in the machine learning and statistics literature but, to the best of our knowledge, have not been used in conjunction with derivative data in the way proposed here. We refer to Minka (2000, Sec. 4); Steinwart and Christmann (2008, Example 4.9) and (Liang and Rakhlin, 2020) for a few of their appearances. Gaussian process regression based on derivative evaluations has been explored (e.g., Solak et al., 2002; Prüher and Särkkä, 2016; Wu et al., 2017; Eriksson et al., 2018), though typically for "standard" kernels such as the Gaussian kernel. The approach in this paper differs from the prior Gaussian process literature in two key ways, which enable a probabilistic replication of the Taylor expansion: First, for kernels used in the literature the posterior mean cannot coincide with the Taylor polynomial. Secondly, in

the literature the data typically consist of function and derivative evaluations at a number of *different points*, whereas we are specifically interested in derivatives at a *single point*.

## 1.2 Contributions

The main contributions of the paper are contained in Sections 2 and 3. In Section 2, we derive a probabilistic Taylor expansion and a basic error bound; these results are given in Theorems 2.1 and 2.3. We also discuss how inclusion of observation noise affects probabilistic Taylor expansions. In Section 3, we derive expressions for maximum likelihood estimates of the Taylor kernel parameters $\sigma$ and $\lambda$. Perhaps the most interesting result that we obtain is Theorem 3.2, which states that derivative data that could have been generated by a constant function yield the estimate $\lambda_{\mathsf{ML}} = 0$. As mentioned above, the exponential kernel is related to the Gaussian kernel. In Section 4, we show how to derive closed form expression for the posterior mean and covariance given derivative data when the covariance kernel is Gaussian. Section 5 outlines generalisations of probabilistic Taylor expansions derived in Section 2 for data that are orthogonal in the reproducing kernel Hilbert space of the covariance kernel. Fourier coefficients constitute an example of orthogonal data when the kernel is periodic. Some simple numerical toy examples are included in Section 6.

We emphasise that the purpose of this paper is not to propose new Gaussian process algorithms but rather to provide a Gaussian process interpretation for classical and well-known Taylor approximations. Such an interpretation is intrinsically interesting even if it yields no new competitive algorithms.

## 1.3 Notation

We use $\mathbb{N}_0$ to denote the set of non-negative integers and $\mathbb{N}_0^d$ to denote the collection of non-negative $d$-dimensional multi-indices $\boldsymbol{\alpha} = (\boldsymbol{\alpha}(1), \ldots, \boldsymbol{\alpha}(d))$, where $\boldsymbol{\alpha}(j) \in \mathbb{N}_0$ is the $j$th index of $\boldsymbol{\alpha}$. We also use the standard notation $|\boldsymbol{\alpha}| = \boldsymbol{\alpha}(1) + \cdots + \boldsymbol{\alpha}(d)$ and $\boldsymbol{\alpha}! = \boldsymbol{\alpha}(1)! \times \cdots \times \boldsymbol{\alpha}(d)!$.

## 2 A Probabilistic Taylor Expansion

In this section, we derive a probabilistic Taylor expansion using Gaussian processes. We discuss a generalisation of this derivation for orthogonal data in Section 5.

### 2.1 Taylor Kernels

Let $\boldsymbol{a} \in \mathbb{R}^d$ and $r \in (0, \infty]$. Define $\Omega_{\boldsymbol{a},r} = \{\boldsymbol{x} \in \mathbb{R}^d : \|\boldsymbol{x} - \boldsymbol{a}\|_2 < r\}$. A *multidimensional Taylor kernel* on $\Omega_{\boldsymbol{a},r} \times \Omega_{\boldsymbol{a},r}$ is defined as

$$K_{\boldsymbol{a}}(\boldsymbol{x}, \boldsymbol{y}) = K(\boldsymbol{x} - \boldsymbol{a}, \boldsymbol{y} - \boldsymbol{a}) \quad \text{for} \quad K(\boldsymbol{x}, \boldsymbol{y}) = \sigma^2 \sum_{\boldsymbol{\alpha} \in \mathbb{N}_0^d} \frac{c_{\boldsymbol{\alpha}} \boldsymbol{\lambda}^{\boldsymbol{\alpha}}}{(\boldsymbol{\alpha}!)^2} \boldsymbol{x}^{\boldsymbol{\alpha}} \boldsymbol{y}^{\boldsymbol{\alpha}}, \tag{2.1}$$

where $\sigma > 0$ and $\boldsymbol{\lambda} \in \mathbb{R}_+^d$ are kernel hyperparameters. The coefficients $c_{\boldsymbol{\alpha}}$ are non-negative constants such that $c_{\boldsymbol{\alpha}} > 0$ for infinitely many $\boldsymbol{\alpha} \in \mathbb{N}_0^d$ and

$$\sum_{\boldsymbol{\alpha} \in \mathbb{N}_0^d} \frac{c_{\boldsymbol{\alpha}} \boldsymbol{\lambda}^{\boldsymbol{\alpha}}}{(\boldsymbol{\alpha}!)^2} r^{2|\boldsymbol{\alpha}|} < \infty \ \text{ if } \ r < \infty \quad \text{or} \quad \sum_{\boldsymbol{\alpha} \in \mathbb{N}_0^d} \frac{c_{\boldsymbol{\alpha}} \boldsymbol{\lambda}^{\boldsymbol{\alpha}}}{\boldsymbol{\alpha}! \sqrt{\boldsymbol{\alpha}!}} \mathrm{e}^{|\boldsymbol{\alpha}|} < \infty \ \text{ if } \ r = \infty. \tag{2.2}$$

The conditions (2.2) are sufficient to ensure the series defining $K_{\boldsymbol{a}}$ via (2.1) converges absolutely for all $\boldsymbol{x}, \boldsymbol{y} \in \Omega_{\boldsymbol{a},r}$, which, together with $c_{\boldsymbol{\alpha}} > 0$ for infinitely many $\boldsymbol{\alpha}$, guarantees that Taylor kernels are positive-definite (Zwicknagl and Schaback, 2013, Thm. 2.2). If only finititely many $c_{\boldsymbol{\alpha}}$ are positive, the kernel is positive-semidefinite. However, $c_{\boldsymbol{\alpha}} \geq 0$ is not necessary for positive-definiteness of $K$ in (2.1) (see Zwicknagl, 2009, Sec. 2). To ensure that the diagonal covariance matrix in (2.11) is invertible we always assume that

$$c_{\boldsymbol{\alpha}} > 0 \quad \text{for every} \quad \boldsymbol{\alpha} \in \mathbb{N}_0^d.$$

Note that $\sigma$ and $\boldsymbol{\lambda}$ could be subsumed into the coefficients $c_{\boldsymbol{\alpha}}$. However, as we shall see in Section 3, the parametrisation that we use leads to convenient and useful hyperparameter estimation. Specifically, maximum

likelihood estimation of the parameters $\sigma$ and $\boldsymbol{\lambda}$ is possible and the estimators have some intuitive properties. In contrast, it is either useless or impossible to estimate the coefficients $c_{\boldsymbol{\alpha}}$, which should therefore be fixed.

An important subclass of Taylor kernels are *inner product kernels*, defined by

$$K(\boldsymbol{x}, \boldsymbol{y}) = \sigma^2 \sum_{p=0}^{\infty} \frac{c_p}{(p!)^2} \langle \boldsymbol{x}, \boldsymbol{y} \rangle_{\boldsymbol{\lambda}}^p, \quad \text{where} \quad \langle \boldsymbol{x}, \boldsymbol{y} \rangle_{\boldsymbol{\lambda}} = \sum_{i=1}^{d} \lambda_i x_i y_i. \tag{2.3}$$

It is easy to show that inner product kernels are Taylor kernels: From the multinomial theorem we have

$$K(\boldsymbol{x}, \boldsymbol{y}) = \sigma^2 \sum_{p=0}^{\infty} \frac{c_p}{(p!)^2} \langle \boldsymbol{x}, \boldsymbol{y} \rangle_{\boldsymbol{\lambda}}^p = \sigma^2 \sum_{p=0}^{\infty} \frac{c_p}{(p!)^2} \left( \sum_{i=1}^{d} \lambda_i x_i y_i \right)^p = \sigma^2 \sum_{p=0}^{\infty} \frac{c_p}{(p!)^2} \sum_{|\boldsymbol{\alpha}|=p} \frac{p!}{\boldsymbol{\alpha}!} \boldsymbol{\lambda}^{\boldsymbol{\alpha}} \boldsymbol{x}^{\boldsymbol{\alpha}} \boldsymbol{y}^{\boldsymbol{\alpha}}$$

$$= \sigma^2 \sum_{\boldsymbol{\alpha} \in \mathbb{N}_0^d} \frac{c_{|\boldsymbol{\alpha}|}}{\boldsymbol{\alpha}! \, |\boldsymbol{\alpha}|!} \boldsymbol{\lambda}^{\boldsymbol{\alpha}} \boldsymbol{x}^{\boldsymbol{\alpha}} \boldsymbol{y}^{\boldsymbol{\alpha}},$$

which we recognise as a Taylor kernel in (2.1) with $c_{\boldsymbol{\alpha}} = c_{|\boldsymbol{\alpha}|} \boldsymbol{\alpha}! / |\boldsymbol{\alpha}|!$. We will discuss estimation of the parameters $\sigma$ and $\boldsymbol{\lambda}$ (as well as the coefficients $c_p$) in Section 3; for now, we assume the parameters are given and proceed to show how Taylor kernels may be used to derive a probabilistic Taylor expansion.

The multidimensional version of the exponential kernel in (1.3) is

$$K(\boldsymbol{x}, \boldsymbol{y}) = \sigma^2 \exp(\langle \boldsymbol{x}, \boldsymbol{y} \rangle_{\boldsymbol{\lambda}}) = \sigma^2 \sum_{p=0}^{\infty} \frac{1}{p!} \langle \boldsymbol{x}, \boldsymbol{y} \rangle_{\boldsymbol{\lambda}}. \tag{2.4}$$

The exponential kernel is defined on $\Omega_{\boldsymbol{a}, r} = \mathbb{R}^d$. In Section 4, we discuss a close connection that the exponential kernel has to the commonly used Gaussian kernel. By setting $c_p = 1$ we obtain the *Bessel kernel* $K(x, y) = \sigma^2 \sum_{p=0}^{\infty} \langle \boldsymbol{x}, \boldsymbol{y} \rangle_{\boldsymbol{\lambda}}^p / (p!)^2 = \mathrm{I}_0(2 \langle \boldsymbol{x}, \boldsymbol{y} \rangle_{\boldsymbol{\lambda}}^{1/2})$, where $\mathrm{I}_0$ is the modified Bessel function of the first kind, which is another Taylor kernel defined on the whole of $\mathbb{R}^d$.

## 2.2 Gaussian Process Regression with Derivative Data

A Gaussian process $f_{\mathsf{GP}} \sim \mathrm{GP}(m, R)$ characterised by mean function $m \colon \Omega_{\boldsymbol{a}, r} \to \mathbb{R}$ and covariance kernel $R \colon \Omega_{\boldsymbol{a}, r} \times \Omega_{\boldsymbol{a}, r} \to \mathbb{R}$ is a stochastic process such that for any points $\boldsymbol{x}_1, \ldots, \boldsymbol{x}_N \in \Omega_{\boldsymbol{a}, r}$ the joint distribution of $(f_{\mathsf{GP}}(\boldsymbol{x}_1), \ldots, f_{\mathsf{GP}}(\boldsymbol{x}_N))$ is an $N$-dimensional Gaussian with mean vector $(m(\boldsymbol{x}_1), \ldots, m(\boldsymbol{x}_N)) \in \mathbb{R}^N$ and covariance matrix $\boldsymbol{R} = (R(\boldsymbol{x}_i, \boldsymbol{x}_j))_{i,j=1}^N \in \mathbb{R}^{N \times N}$ (Rasmussen and Williams, 2006). In particular, $\mathbb{E}[f_{\mathsf{GP}}(\boldsymbol{x})] = m(\boldsymbol{x})$ and $\mathrm{Cov}[f(\boldsymbol{x}), f(\boldsymbol{y})] = R(\boldsymbol{x}, \boldsymbol{y})$ for all $\boldsymbol{x}, \boldsymbol{y} \in \Omega_{\boldsymbol{a}, r}$. Let $f \colon \Omega_{\boldsymbol{a}, r} \to \mathbb{R}$ be an $n$ times differentiable function on $\Omega_{\boldsymbol{a}, r}$, meaning that the partial derivatives

$$\mathrm{D}^{\boldsymbol{\alpha}} f = \frac{\partial^{|\boldsymbol{\alpha}|} f}{\partial x_1^{\boldsymbol{\alpha}(1)} \cdots \partial x_d^{\boldsymbol{\alpha}(d)}}$$

exist for all $\boldsymbol{\alpha} \in \mathbb{N}_0^d$ such that $|\boldsymbol{\alpha}| \leq n$. Suppose also that the prior mean $m$ is $n$ times differentiable and that $R$ is $n$ times differentiable in both arguments, in the sense that the derivative

$$\mathrm{D}_{\boldsymbol{y}}^{\boldsymbol{\beta}} \mathrm{D}_{\boldsymbol{x}}^{\boldsymbol{\alpha}} R(\boldsymbol{x}, \boldsymbol{y}) = \frac{\partial^{|\boldsymbol{\alpha}| + |\boldsymbol{\beta}|}}{\partial \boldsymbol{v}^{\boldsymbol{\alpha}} \partial \boldsymbol{w}^{\boldsymbol{\beta}}} R(\boldsymbol{v}, \boldsymbol{w}) \bigg|_{\substack{\boldsymbol{v} = \boldsymbol{x} \\ \boldsymbol{w} = \boldsymbol{y}}}$$

exists for all $\boldsymbol{x}, \boldsymbol{y} \in \Omega_{\boldsymbol{a}, r}$ and all multi-indices $\boldsymbol{\alpha}$ and $\boldsymbol{\beta}$ such that $|\boldsymbol{\alpha}|, |\boldsymbol{\beta}| \leq n$. The *noiseless* derivative data are

$$\boldsymbol{f_a} = (\mathrm{D}^{\boldsymbol{\alpha}} f(\boldsymbol{a}))_{|\boldsymbol{\alpha}| \leq n} = (\mathrm{D}^{\boldsymbol{\alpha}_1} f(\boldsymbol{a}), \ldots, \mathrm{D}^{\boldsymbol{\alpha}_{N_n^d}} f(\boldsymbol{a})), \tag{2.5}$$

where we use an arbitrary ordering of the set $\{\boldsymbol{\alpha}_1, \ldots, \boldsymbol{\alpha}_{N_n^d}\} = \{\boldsymbol{\alpha} \in \mathbb{N}_0^d : |\boldsymbol{\alpha}| \leq n\}$, which contains

$$N_n^d = \binom{n+d}{n} = \frac{(n+d)!}{n! \, d!}$$

elements. When conditioned on these derivative data, the posterior $f_{\mathsf{GP}} \mid \boldsymbol{f_a}$ is a Gaussian process (Särkkä, 2011; Travelletti and Ginsbourger, 2022). That is, $f_{\mathsf{GP}} \mid \boldsymbol{f_a} \sim \mathrm{GP}(s_{n,\boldsymbol{a}}, P_{n,\boldsymbol{a}})$ with mean and covariance

$$s_{n,\boldsymbol{a}}(\boldsymbol{x}) = m(\boldsymbol{x}) + \boldsymbol{r_a}(\boldsymbol{x})^\mathsf{T} \boldsymbol{R_a}^{-1}(\boldsymbol{f_a} - \boldsymbol{m_a}) \quad \text{and} \quad P_{n,\boldsymbol{a}}(\boldsymbol{x}, \boldsymbol{y}) = R(\boldsymbol{x}, \boldsymbol{y}) - \boldsymbol{r_a}(\boldsymbol{x})^\mathsf{T} \boldsymbol{R_a}^{-1} \boldsymbol{r_a}(\boldsymbol{y}). \qquad (2.6)$$

Here $\boldsymbol{R_a} \in \mathbb{R}^{N_n^d \times N_n^d}$ and $\boldsymbol{r_a}(\boldsymbol{x}) \in \mathbb{R}^{N_n^d}$ are each given by

$$\left(\boldsymbol{R_a}\right)_{ij} = \mathrm{D}_{\boldsymbol{y}}^{\boldsymbol{\alpha}_j} \mathrm{D}_{\boldsymbol{x}}^{\boldsymbol{\alpha}_i} R(\boldsymbol{x}, \boldsymbol{y})\big|_{\substack{\boldsymbol{x}=\boldsymbol{a} \\ \boldsymbol{y}=\boldsymbol{a}}} \quad \text{and} \quad \left(\boldsymbol{r_a}(\boldsymbol{x})\right)_i = \mathrm{D}_{\boldsymbol{y}}^{\boldsymbol{\alpha}_i} R(\boldsymbol{x}, \boldsymbol{y})\big|_{\boldsymbol{y}=\boldsymbol{a}}, \qquad (2.7)$$

where subscripts denote the differentiation variable, and $\boldsymbol{m_a} = (\mathrm{D}^{\boldsymbol{\alpha}_1} m(\boldsymbol{a}), \dots, \mathrm{D}^{\boldsymbol{\alpha}_{N_n^d}} m(\boldsymbol{a}))$. When $f$ has multidimensional range, one may model each of its components independently, though we note that this modelling choice may be readily generalised using vector-valued Gaussian processes (Álvarez et al., 2012).

## 2.3 Replicating the Taylor Expansion Using Taylor Kernels

The next theorem combines what was described in Sections 2.1 and 2.2 to give a probabilistic version of the Taylor expansion.

**Theorem 2.1.** *Let $K_{\boldsymbol{a}}$ be a Taylor kernel defined as in* (2.1). *Let $f_{\mathsf{GP}} \sim \mathrm{GP}(m, K_{\boldsymbol{a}})$ and $\boldsymbol{f_a} = (\mathrm{D}^{\boldsymbol{\alpha}} f(\boldsymbol{a}))_{|\boldsymbol{\alpha}| \le n}$. Then $f_{\mathsf{GP}} \mid \boldsymbol{f_a} \sim \mathrm{GP}(s_{n,\boldsymbol{a}}, P_{n,\boldsymbol{a}})$, where*

$$s_{n,\boldsymbol{a}}(\boldsymbol{x}) = m(\boldsymbol{x}) + \sum_{|\boldsymbol{\alpha}| \le n} \frac{\mathrm{D}^{\boldsymbol{\alpha}}[f(\boldsymbol{a}) - m(\boldsymbol{a})]}{\boldsymbol{\alpha}!} (\boldsymbol{x} - \boldsymbol{a})^{\boldsymbol{\alpha}} \ \text{and} \ P_{n,\boldsymbol{a}}(\boldsymbol{x}, \boldsymbol{y}) = \sigma^2 \sum_{|\boldsymbol{\alpha}| > n} \frac{c_{\boldsymbol{\alpha}} \boldsymbol{\lambda}^{\boldsymbol{\alpha}}}{(\boldsymbol{\alpha}!)^2} (\boldsymbol{x} - \boldsymbol{a})^{\boldsymbol{\alpha}} (\boldsymbol{y} - \boldsymbol{a})^{\boldsymbol{\alpha}}. \ (2.8)$$

*If $m$ is a polynomial of degree at most $n$, then*

$$s_{n,\boldsymbol{a}}(\boldsymbol{x}) = \sum_{|\boldsymbol{\alpha}| \le n} \frac{\mathrm{D}^{\boldsymbol{\alpha}} f(\boldsymbol{a})}{\boldsymbol{\alpha}!} (\boldsymbol{x} - \boldsymbol{a})^{\boldsymbol{\alpha}}, \qquad (2.9)$$

*which is identical to the multidimensional version of the Taylor polynomial in* (1.1).

*Proof.* It is straightforward to compute that, for any $\boldsymbol{\beta}, \boldsymbol{\gamma} \in \mathbb{N}_0^d$,

$$\mathrm{D}_{\boldsymbol{y}}^{\boldsymbol{\beta}} \mathrm{D}_{\boldsymbol{x}}^{\boldsymbol{\gamma}} K_{\boldsymbol{a}}(\boldsymbol{x}, \boldsymbol{y}) = \sigma^2 \sum_{\boldsymbol{\alpha} \ge \boldsymbol{\beta} \wedge \boldsymbol{\gamma}} \frac{c_{\boldsymbol{\alpha}} \boldsymbol{\lambda}^{\boldsymbol{\alpha}} (\boldsymbol{x} - \boldsymbol{a})^{\boldsymbol{\alpha} - \boldsymbol{\beta}} (\boldsymbol{y} - \boldsymbol{a})^{\boldsymbol{\alpha} - \boldsymbol{\gamma}}}{(\boldsymbol{\alpha} - \boldsymbol{\beta})! (\boldsymbol{\alpha} - \boldsymbol{\gamma})!}, \qquad (2.10)$$

where $\boldsymbol{\beta} \wedge \boldsymbol{\gamma} = (\max\{\boldsymbol{\beta}(1), \boldsymbol{\gamma}(1)\}, \dots, \max\{\boldsymbol{\beta}(d), \boldsymbol{\gamma}(d)\})$. If $\boldsymbol{x} = \boldsymbol{a}$ or $\boldsymbol{y} = \boldsymbol{a}$, all terms with $\boldsymbol{\alpha} - \boldsymbol{\beta} \ne \boldsymbol{0}$ or $\boldsymbol{\alpha} - \boldsymbol{\gamma} \ne \boldsymbol{0}$, respectively, in (2.10) vanish. Therefore in the context of (2.7) we have

$$\left(\boldsymbol{R_a}\right)_{ij} = \sigma^2 c_{\boldsymbol{\alpha}_i} \boldsymbol{\lambda}^{\boldsymbol{\alpha}_i} \delta_{ij} \quad \text{and} \quad \left(\boldsymbol{r_a}(\boldsymbol{x})\right)_i = \sigma^2 \frac{c_{\boldsymbol{\alpha}_i} \boldsymbol{\lambda}^{\boldsymbol{\alpha}_i}}{\boldsymbol{\alpha}_i!} (\boldsymbol{x} - \boldsymbol{a})^{\boldsymbol{\alpha}_i}. \qquad (2.11)$$

Consequently, the matrix $\boldsymbol{R_a}$ is diagonal and the $i$th element of the row vector $\boldsymbol{r_a}(\boldsymbol{x})^\mathsf{T} \boldsymbol{R_a}^{-1}$ in (2.6) is $(\boldsymbol{\alpha}_i!)^{-1}(\boldsymbol{x} - \boldsymbol{a})^{\boldsymbol{\alpha}_i}$. It follows that the posterior mean and covariance are as in (2.8). From (2.1) we recognise the covariance $P_{n,\boldsymbol{a}}$ as the remainder in the kernel expansion. To prove (2.9) it is sufficient to observe that $m(\boldsymbol{x}) = \sum_{|\boldsymbol{\alpha}| \le n} \mathrm{D}^{\boldsymbol{\alpha}} m(\boldsymbol{a})(\boldsymbol{\alpha}!)^{-1}(\boldsymbol{x} - \boldsymbol{a})^{\boldsymbol{\alpha}}$ when $m$ is a polynomial of degree at most $n$. By inspection it is clear that $s_{n,\boldsymbol{a}}$ in (2.9) is identical to the Taylor expansion given in (1.1) (and its multivariate version), which completes the proof. $\qquad \square$

Note that the covariance is not identically zero—in fact, $P_{n,\boldsymbol{a}}(\boldsymbol{x}, \boldsymbol{x}) \to \infty$ as $\|\boldsymbol{x} - \boldsymbol{a}\|_2 \to \infty$. Furthermore, while $P_{n,\boldsymbol{a}}(\boldsymbol{x}, \boldsymbol{y})$ takes the form of an infinite sum, provided $K_{\boldsymbol{a}}$ has a closed form it can be computed by subtracting the terms with $|\boldsymbol{\alpha}| \le n$ in the summation form of $K_{\boldsymbol{a}}$ from that closed form. For illustration, some posterior processes are displayed in Figure 1.

Whether or not the explosion of the posterior variance away from $\boldsymbol{a}$ is desirable depends on what one is trying to achieve and what kind of prior information is available. If one is trying to extrapolate, it seems

entirely natural to us, at least in the absence of additional knowledge about $f$, that the variance should be very large far away from $\boldsymbol{a}$. But if there is additional prior information that the function $f$ has, for example, approximately the same magnitude everywhere on its domain, then it may make sense to use a stationary kernel for which the variance tends to a constant value as the distance to the nearest data point increases.

The expressions in (2.8) for the posterior mean and covariance show that computational complexity of inference with Taylor kernels and derivative data is linear in the number of data points, $N_n^d$, if the derivatives of $m$ are cheap to compute (e.g., if $m$ is a polynomial). A generic kernel for which no special structure is present in the covariance matrix $\boldsymbol{R_a}$ incurs cubic computational cost because a linear system of equations needs to solved when the mean and variance are computed directly from (2.6). This seeming advantage of Taylor kernels is lost if the data do not consist of derivatives at a single point.

**Remark 2.2.** Recall that in Section 2.1 we assumed that $c_{\boldsymbol{\alpha}} > 0$ for every $\boldsymbol{\alpha} \in \mathbb{N}_0^d$. However, from (2.11) we easily see that Theorem 2.1 remains valid as long as $c_{\boldsymbol{\alpha}} > 0$ for all $|\boldsymbol{\alpha}| \leq n$ because this ensures that the diagonal covariance matrix $\boldsymbol{R_a}$ is invertible. Section 2.1 therefore applies also to *polynomial kernels*, which are Taylor kernels with finitely many non-zero coefficients $c_{\boldsymbol{\alpha}}$ if $n$ remains sufficiently small.

## 2.4 Error Bounds

Each positive-semidefinite kernel $R$ on $\Omega_{\boldsymbol{a},r} \times \Omega_{\boldsymbol{a},r}$ is associated to a unique *reproducing kernel Hilbert space* (RKHS), $\mathcal{H}(R)$, equipped with inner product $\langle \cdot, \cdot \rangle_{\mathcal{H}(R)}$ and norm $\|\cdot\|_{\mathcal{H}(R)}$. The RKHS is a Hilbert space of functions $f \colon \Omega_{\boldsymbol{a},r} \to \mathbb{R}$ such that $R(\cdot, \boldsymbol{x}) \in \mathcal{H}(R)$ for every $\boldsymbol{x} \in \Omega_{\boldsymbol{a},r}$ and in which the kernel $R$ has the *reproducing property*

$$f(\boldsymbol{x}) = \langle f, R(\cdot, \boldsymbol{x}) \rangle_{\mathcal{H}(R)} \quad \text{for all} \quad f \in \mathcal{H}(R) \text{ and } \boldsymbol{x} \in \Omega_{\boldsymbol{a},r}.$$

See Berlinet and Thomas-Agnan (2004) for more information on RKHSs. It is often difficult to characterise the functions which lie in the RKHS. Fortunately, for Taylor kernels one may use results such as Theorem 9 in Minh (2010) to show that

$$\mathcal{H}(K_{\boldsymbol{a}}) = \left\{ f(\boldsymbol{x}) = \sigma \sum_{\boldsymbol{\alpha} \in \mathbb{N}_0^d} f_{\boldsymbol{\alpha}} \frac{\sqrt{c_{\boldsymbol{\alpha}} \boldsymbol{\lambda}^{\boldsymbol{\alpha}}}}{\boldsymbol{\alpha}!} (\boldsymbol{x} - \boldsymbol{a})^{\boldsymbol{\alpha}} \; : \; \|f\|_{\mathcal{H}(K_{\boldsymbol{a}})}^2 = \sum_{\boldsymbol{\alpha} \in \mathbb{N}_0^d} f_{\boldsymbol{\alpha}}^2 < \infty \right\}.$$

See also Zwicknagl and Schaback (2013) and Paulsen and Raghupathi (2016, Sec. 2.1). For example, all polynomials are contained in the RKHS of any Taylor kernel for any $\boldsymbol{a} \in \mathbb{R}^d$.

The next theorem shows that the posterior variance has a similar interpretation to the Taylor remainder term if $f$ is in $\mathcal{H}(K_{\boldsymbol{a}})$.

**Theorem 2.3.** *Let $f_{\mathsf{GP}} \mid \boldsymbol{f_a}$ be as in Theorem 2.1, and let the assumptions of that theorem hold. If $f \in \mathcal{H}(K_{\boldsymbol{a}})$, then $s_{n,\boldsymbol{a}}$ and $P_{n,\boldsymbol{a}}$ satisfy*

$$|f(\boldsymbol{x}) - s_{n,\boldsymbol{a}}(\boldsymbol{x})| \leq \|f\|_{\mathcal{H}(K_{\boldsymbol{a}})} P_{n,\boldsymbol{a}}(\boldsymbol{x}, \boldsymbol{x})^{1/2} \leq C_{n,r} \, \sigma \, \|f\|_{\mathcal{H}(K_{\boldsymbol{a}})} \|\boldsymbol{x} - \boldsymbol{a}\|_2^{n+1} \tag{2.12}$$

*for all $\boldsymbol{x} \in \Omega_{\boldsymbol{a},r}$, where $(C_{n,r})_{n=0}^{\infty}$ is a positive sequence such that $C_{n,r} \to 0$ as $n \to \infty$.*

*Proof.* By the standard equivalence between Gaussian process regression in the noiseless setting and worst-case optimal approximation (e.g., Scheuerer et al., 2013; Kanagawa et al., 2018) the posterior mean $s_{n,\boldsymbol{a}} \in \mathcal{H}(K_{\boldsymbol{a}})$ is the minimum-norm approximant of $f$ such that $\mathrm{D}^{\boldsymbol{\alpha}} s_{n,\boldsymbol{a}}(\boldsymbol{a}) = \mathrm{D}^{\boldsymbol{\alpha}} f(\boldsymbol{a})$ for every $|\boldsymbol{\alpha}| \leq n$ and the posterior standard deviation at $\boldsymbol{x}$, $P_{n,\boldsymbol{a}}(\boldsymbol{x}, \boldsymbol{x})^{1/2}$, equals the worst-case approximation error at $\boldsymbol{x}$ in the RKHS. Hence

$$|f(\boldsymbol{x}) - s_{n,\boldsymbol{a}}(\boldsymbol{x})| \leq \|f - s_{\boldsymbol{a}}\|_{\mathcal{H}(K_{\boldsymbol{a}})} P_{n,\boldsymbol{a}}(\boldsymbol{x}, \boldsymbol{x})^{1/2} \leq \|f\|_{\mathcal{H}(K_{\boldsymbol{a}})} P_{n,\boldsymbol{a}}(\boldsymbol{x}, \boldsymbol{x})^{1/2}$$

for all $\boldsymbol{x} \in \Omega_{\boldsymbol{a},r}$ if $f \in \mathcal{H}(K_{\boldsymbol{a}})$ (e.g., Wendland, 2005, Thm. 16.3). To prove the upper bound for the posterior variance observe that from the general inequality $|\boldsymbol{x}^{\boldsymbol{\alpha}}| \leq \|\boldsymbol{x}\|_{\infty}^{|\boldsymbol{\alpha}|} \leq \|\boldsymbol{x}\|_2^{|\boldsymbol{\alpha}|}$ for $\boldsymbol{x} \in \mathbb{R}^d$ and $\boldsymbol{\alpha} \in \mathbb{N}_0^d$ it follows

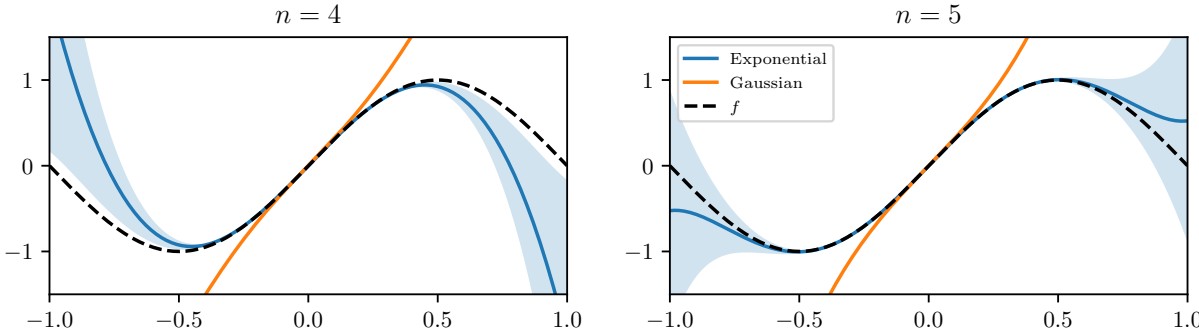

Figure 1: Gaussian process posterior means and 95% credible intervals given derivative data for $f(x) = \sin(\pi x)$ at $a = 0$. The priors have $m \equiv 0$ and use either (blue) the exponential kernel $K(x, y) = \sigma^2 \exp(\lambda xy)$ with $\lambda = 3/2$ or (orange) the Gaussian kernel $K(x, y) = \sigma^2 \exp(-\lambda^2(x - y)^2/2)$. Maximum likelihood estimation was used to select the scaling parameter $\sigma$. We shall revisit this example in Section 6.

that, for any $\boldsymbol{x} \in \Omega_{\boldsymbol{a},r} = \{\boldsymbol{x} \in \mathbb{R}^d : \|\boldsymbol{x} - \boldsymbol{a}\|_2 < r\}$,

$$
\begin{aligned}
P_{n,\boldsymbol{a}}(\boldsymbol{x}, \boldsymbol{x}) &= \sigma^2 \sum_{|\boldsymbol{\alpha}| > n} \frac{c_{\boldsymbol{\alpha}} \boldsymbol{\lambda}^{\boldsymbol{\alpha}}}{(\boldsymbol{\alpha}!)^2} (\boldsymbol{x} - \boldsymbol{a})^{2\boldsymbol{\alpha}} \\
&= \sigma^2 \left( \sum_{|\boldsymbol{\alpha}| = n+1} \frac{c_{\boldsymbol{\alpha}} \boldsymbol{\lambda}^{\boldsymbol{\alpha}}}{(\boldsymbol{\alpha}!)^2} (\boldsymbol{x} - \boldsymbol{a})^{2\boldsymbol{\alpha}} + \sum_{|\boldsymbol{\alpha}| > n+1} \frac{c_{\boldsymbol{\alpha}} \boldsymbol{\lambda}^{\boldsymbol{\alpha}}}{(\boldsymbol{\alpha}!)^2} (\boldsymbol{x} - \boldsymbol{a})^{2\boldsymbol{\alpha}} \right) \\
&\leq \sigma^2 \left( \sum_{|\boldsymbol{\alpha}| = n+1} \frac{c_{\boldsymbol{\alpha}} \boldsymbol{\lambda}^{\boldsymbol{\alpha}}}{(\boldsymbol{\alpha}!)^2} \|\boldsymbol{x} - \boldsymbol{a}\|_2^{2(n+1)} + \sum_{|\boldsymbol{\alpha}| > n+1} \frac{c_{\boldsymbol{\alpha}} \boldsymbol{\lambda}^{\boldsymbol{\alpha}}}{(\boldsymbol{\alpha}!)^2} \|\boldsymbol{x} - \boldsymbol{a}\|_2^{2|\boldsymbol{\alpha}|} \right) \\
&= \sigma^2 \|\boldsymbol{x} - \boldsymbol{a}\|_2^{2(n+1)} \left( \sum_{|\boldsymbol{\alpha}| = n+1} \frac{c_{\boldsymbol{\alpha}} \boldsymbol{\lambda}^{\boldsymbol{\alpha}}}{(\boldsymbol{\alpha}!)^2} + \sum_{|\boldsymbol{\alpha}| > n+1} \frac{c_{\boldsymbol{\alpha}} \boldsymbol{\lambda}^{\boldsymbol{\alpha}}}{(\boldsymbol{\alpha}!)^2} \|\boldsymbol{x} - \boldsymbol{a}\|_2^{2|\boldsymbol{\alpha}| - 2(n+1)} \right) \\
&\leq \sigma^2 \|\boldsymbol{x} - \boldsymbol{a}\|_2^{2(n+1)} \underbrace{\left( \sum_{|\boldsymbol{\alpha}| = n+1} \frac{c_{\boldsymbol{\alpha}} \boldsymbol{\lambda}^{\boldsymbol{\alpha}}}{(\boldsymbol{\alpha}!)^2} + r^{-2(n+1)} \sum_{|\boldsymbol{\alpha}| > n+1} \frac{c_{\boldsymbol{\alpha}} \boldsymbol{\lambda}^{\boldsymbol{\alpha}}}{(\boldsymbol{\alpha}!)^2} r^{2|\boldsymbol{\alpha}|} \right)}_{=C_{n,r}}.
\end{aligned}
$$

The summability assumption (2.2) ensures that $C_{n,r}$ is finite and that $C_{n,r} \to 0$ as $n \to \infty$. □

The bound (2.12) is valid for every element of $\mathcal{H}(K_{\boldsymbol{a}})$. However, the bound is *uncomputable* because it is not possible to compute the norm $\|f\|_{\mathcal{H}(K_{\boldsymbol{a}})}$ from the derivative data $\boldsymbol{f_a}$ without some additional information about the function $f$. For example, when $d = 1$ and $a = 0$, the functions

$$
f_1(x) = 1 + 2x + 3x^2 + 4x^3 + 5x^4 \quad \text{and} \quad f_2(x) = 1 + 2x + 3x^2 + 4x^3 + cx^7
$$

are different if $c \neq 0$ but provide the same derivative data whenever $n \leq 3$. In practice, what one has to do is estimate the parameters of $K_{\boldsymbol{a}}$ in a data-dependent way, use the standard deviation $P_{n,\boldsymbol{a}}(\boldsymbol{x}, \boldsymbol{x})^{1/2}$ to compute, say, the 95% credible interval around the point estimate $s_{n,\boldsymbol{a}}(\boldsymbol{x})$ of $f(\boldsymbol{x})$, and conclude that it is likely that $f(\boldsymbol{x})$ falls within the resulting credible interval. Any such uncertainty estimates are bound to fail occasionally—and the severity of the failure can be arbitrary. For example, when $n = 3$, credible intervals formed from derivative data generated the function $f_2$ above do not depend on $c$ even though $|f_2(x) - s_{3,0}(x)| = cx^7$ does.

### 2.5 Noisy Observations

Suppose that the observations are corrupted by Gaussian noise. That is, the data vector is $\boldsymbol{f_a} = (y_{\boldsymbol{\alpha}})_{|\boldsymbol{\alpha}| \leq n}$, where $y_{\boldsymbol{\alpha}} = \mathrm{D}^{\boldsymbol{\alpha}} f(\boldsymbol{a}) + z_{\boldsymbol{\alpha}}$ with independent $z_{\boldsymbol{\alpha}} \sim \mathrm{N}(0, \varepsilon_{\boldsymbol{\alpha}}^2)$ for $\varepsilon_{\boldsymbol{\alpha}} > 0$. While unrealistic in practice, the assumption that the noise terms are independent allows for explicit computation of the posterior mean and variance. In this setting the posterior mean and covariance for a general sufficiently differentiable prior mean $m$ and covariance kernel $R$ are

$$s_{n,\boldsymbol{a}}(\boldsymbol{x}) = m(\boldsymbol{x}) + \boldsymbol{r_a}(\boldsymbol{x})^{\mathsf{T}}(\boldsymbol{R_a} + \boldsymbol{E})^{-1}(\boldsymbol{f_a} - \boldsymbol{m_a}) \text{ and } P_{n,\boldsymbol{a}}(\boldsymbol{x}, \boldsymbol{y}) = R(\boldsymbol{x}, \boldsymbol{y}) - \boldsymbol{r_a}(\boldsymbol{x})^{\mathsf{T}}(\boldsymbol{R_a} + \boldsymbol{E})^{-1}\boldsymbol{r_a}(\boldsymbol{y}), \quad (2.13)$$

where $\boldsymbol{E}$ is a diagonal $N_n^d \times N_n^d$ matrix containing the noise variances $\varepsilon_{\boldsymbol{a}}^2$ and $\boldsymbol{r_a}(\boldsymbol{x})$, $\boldsymbol{R_a}$, and $\boldsymbol{m_a}$ were defined in Section 2.2. Recall then from Section 2.3 that when $R$ is a Taylor kernel $K_{\boldsymbol{a}}$ we have

$$(\boldsymbol{R_a})_{ij} = \sigma^2 c_{\boldsymbol{\alpha}_i} \boldsymbol{\lambda}^{\boldsymbol{\alpha}_i} \delta_{ij} \quad \text{and} \quad (\boldsymbol{r_a}(\boldsymbol{x}))_i = \sigma^2 \frac{c_{\boldsymbol{\alpha}_i} \boldsymbol{\lambda}^{\boldsymbol{\alpha}_i}}{\boldsymbol{\alpha}_i!}(\boldsymbol{x} - \boldsymbol{a})^{\boldsymbol{\alpha}_i}.$$

Therefore $(\boldsymbol{R_a} + \boldsymbol{E})^{-1} = (\sigma^2 c_{\boldsymbol{\alpha}_i} \boldsymbol{\lambda}^{\boldsymbol{\alpha}_i} + \varepsilon_{\boldsymbol{\alpha}_i}^2)^{-1}\delta_{ij}$, and plugging this into (2.13) yields

$$s_{n,\boldsymbol{a}}(\boldsymbol{x}) = m(\boldsymbol{x}) + \sigma^2 \sum_{|\boldsymbol{\alpha}| \leq n} \frac{c_{\boldsymbol{\alpha}} \boldsymbol{\lambda}^{\boldsymbol{\alpha}} [y_{\boldsymbol{\alpha}} - \mathrm{D}^{\boldsymbol{\alpha}} m(\boldsymbol{a})]}{\boldsymbol{\alpha}!(\sigma^2 c_{\boldsymbol{\alpha}} \boldsymbol{\lambda}^{\boldsymbol{\alpha}} + \varepsilon_{\boldsymbol{\alpha}}^2)}(\boldsymbol{x} - \boldsymbol{a})^{\boldsymbol{\alpha}}$$

and

$$P_{n,\boldsymbol{a}}(\boldsymbol{x}, \boldsymbol{y}) = K_{\boldsymbol{a}}(\boldsymbol{x}, \boldsymbol{y}) - \sigma^2 \sum_{|\boldsymbol{\alpha}| \leq n} \frac{\sigma^2 c_{\boldsymbol{\alpha}}^2 \boldsymbol{\lambda}^{2\boldsymbol{\alpha}}}{(\boldsymbol{\alpha}!)^2 (\sigma^2 c_{\boldsymbol{\alpha}} \boldsymbol{\lambda}^{\boldsymbol{\alpha}} + \varepsilon_{\boldsymbol{\alpha}}^2)}(\boldsymbol{x} - \boldsymbol{a})^{\boldsymbol{\alpha}}(\boldsymbol{y} - \boldsymbol{a})^{\boldsymbol{\alpha}}$$

$$= \sigma^2 \left[ \sum_{\boldsymbol{\alpha} \in \mathbb{N}_0^d} \frac{c_{\boldsymbol{\alpha}} \boldsymbol{\lambda}^{\boldsymbol{\alpha}}}{(\boldsymbol{\alpha}!)^2}(\boldsymbol{x} - \boldsymbol{a})^{\boldsymbol{\alpha}}(\boldsymbol{y} - \boldsymbol{a})^{\boldsymbol{\alpha}} - \sum_{|\boldsymbol{\alpha}| \leq n} \frac{\sigma^2 c_{\boldsymbol{\alpha}}^2 \boldsymbol{\lambda}^{2\boldsymbol{\alpha}}}{(\boldsymbol{\alpha}!)^2 (\sigma^2 c_{\boldsymbol{\alpha}} \boldsymbol{\lambda}^{\boldsymbol{\alpha}} + \varepsilon_{\boldsymbol{\alpha}}^2)}(\boldsymbol{x} - \boldsymbol{a})^{\boldsymbol{\alpha}}(\boldsymbol{y} - \boldsymbol{a})^{\boldsymbol{\alpha}} \right]$$

$$= \sigma^2 \left[ \sum_{|\boldsymbol{\alpha}| \leq n} \frac{c_{\boldsymbol{\alpha}} \boldsymbol{\lambda}^{\boldsymbol{\alpha}} \varepsilon_{\boldsymbol{\alpha}}^2}{(\boldsymbol{\alpha}!)^2 (\sigma^2 c_{\boldsymbol{\alpha}} \boldsymbol{\lambda}^{\boldsymbol{\alpha}} + \varepsilon_{\boldsymbol{\alpha}}^2)}(\boldsymbol{x} - \boldsymbol{a})^{\boldsymbol{\alpha}}(\boldsymbol{y} - \boldsymbol{a})^{\boldsymbol{\alpha}} + \sum_{|\boldsymbol{\alpha}| > n} \frac{c_{\boldsymbol{\alpha}} \boldsymbol{\lambda}^{\boldsymbol{\alpha}}}{(\boldsymbol{\alpha}!)^2}(\boldsymbol{x} - \boldsymbol{a})^{\boldsymbol{\alpha}}(\boldsymbol{y} - \boldsymbol{a})^{\boldsymbol{\alpha}} \right].$$

Note that by setting $\varepsilon_{\boldsymbol{\alpha}} = 0$ for every $\boldsymbol{\alpha} \in \mathbb{N}_0^d$ such that $|\boldsymbol{\alpha}| \leq n$ we recover the noiseless posterior mean and covariance in (2.8).

## 3 Parameter Estimation

Observe from (2.8) that, although they do not affect the posterior mean, proper selection of the Taylor kernel parameters $\boldsymbol{\lambda}$ and $\sigma$ is a prerequisite for useful and meaningful uncertainty quantification via the posterior variance $P_{n,\boldsymbol{a}}(\boldsymbol{x}, \boldsymbol{x})$. In this section we consider maximum likelihood estimation of these parameters. For the Gaussian process model in Section 2.2, the negative log-likelihood function that is to be minimised with respect to a generic vector of kernel hyperparameters $\boldsymbol{\theta}$ is

$$\ell(\boldsymbol{\theta}) = \frac{1}{2}(\boldsymbol{f_a} - \boldsymbol{m_a})^{\mathsf{T}} \boldsymbol{R_a}^{-1}(\boldsymbol{f_a} - \boldsymbol{m_a}) + \frac{1}{2} \log \det \boldsymbol{R_a} + \frac{N_n^d}{2} \log(2\pi).$$

By discarding terms and coefficients that do not depend on $\theta$ and using (2.11) we see that for a Taylor kernel the maximum likelihood estimate (MLE) $\boldsymbol{\theta}_{\mathsf{ML}}$ is any minimiser of the function

$$\tilde{\ell}(\boldsymbol{\theta}) = \frac{1}{\sigma^2} \sum_{|\boldsymbol{\alpha}| \leq n} \frac{(\mathrm{D}^{\boldsymbol{\alpha}}[f(\boldsymbol{a}) - m(\boldsymbol{a})])^2}{c_{\boldsymbol{\alpha}} \boldsymbol{\lambda}^{\boldsymbol{\alpha}}} + N_n^d \log \sigma^2 + \sum_{|\boldsymbol{\alpha}| \leq n} \log \boldsymbol{\lambda}^{\boldsymbol{\alpha}} + \sum_{|\boldsymbol{\alpha}| \leq n} c_{\boldsymbol{\alpha}}. \quad (3.1)$$

In principle, every coefficient $c_{\boldsymbol{\alpha}}$ of a Taylor kernel in (2.1) may be considered a free parameter to be estimated. However, maximum likelihood estimation of these coefficient is either useless or impossible: From (2.8) we see that the posterior process depends on $c_{\boldsymbol{\alpha}}$ only for $|\boldsymbol{\alpha}| > n$. However the objective function (3.1) does not depend on these $c_{\boldsymbol{\alpha}}$, making it impossible to estimate those parameters that actually influence posterior uncertainty. We encountered a simple example of this phenomenon in Section 2.4.

### 3.1 Estimation of $\sigma$

From (3.1) it is easy to calculate $\sigma_{\mathsf{ML}}$, the maximum likelihood estimate of $\sigma$, for any fixed $\boldsymbol{\lambda} \in \mathbb{R}^d_+$ and $n \in \mathbb{N}_0$ by differentiating $\tilde{\ell}$ and setting its derivative to zero. This gives

$$\sigma^2_{\mathsf{ML}} = \frac{1}{N^d_n} \sum_{|\boldsymbol{\alpha}| \leq n} \frac{(\mathrm{D}^{\boldsymbol{\alpha}}[f(\boldsymbol{a}) - m(\boldsymbol{a})])^2}{c_{\boldsymbol{\alpha}} \boldsymbol{\lambda}^{\boldsymbol{\alpha}}}. \tag{3.2}$$

### 3.2 Estimation of $\boldsymbol{\lambda}$

Estimation of $\boldsymbol{\lambda}$ for a fixed $\sigma$ is more complicated and the maximum likelihood estimate does not appear to admit a closed form expression akin to that for $\sigma_{\mathsf{ML}}$ in (3.2). However, something interesting can be said. We write $\boldsymbol{\lambda} = (\lambda_1, \ldots, \lambda_d)$.

**Lemma 3.1.** *Suppose that $n \geq 1$ and let $1 \leq i \leq d$. Then $\lim_{\lambda_i \to 0} \tilde{\ell}(\boldsymbol{\lambda}) = -\infty$ if $\mathrm{D}^{\boldsymbol{\alpha}}[f(\boldsymbol{a}) - m(\boldsymbol{a})] = 0$ for every $|\boldsymbol{\alpha}| \leq n$ such that $\boldsymbol{\alpha}(i) > 0$ and $\lim_{\lambda_i \to 0} \tilde{\ell}(\boldsymbol{\lambda}) = \infty$ otherwise.*

*Proof.* Assume first that $\mathrm{D}^{\boldsymbol{\alpha}}[f(\boldsymbol{a}) - m(\boldsymbol{a})] = 0$ for every $|\boldsymbol{\alpha}| \leq n$ such that $\boldsymbol{\alpha}(i) > 0$. It follows that the first term in (3.1) does not depend on $\lambda_i$. Because $\sum_{|\boldsymbol{\alpha}| \leq n} \log \boldsymbol{\lambda}^{\boldsymbol{\alpha}} \to -\infty$ as $\lambda_i \to 0$, we have $\tilde{\ell}(\boldsymbol{\lambda}) \to -\infty$ as $\lambda_i \to 0$. Assume then that $\mathrm{D}^{\boldsymbol{\beta}}[f(\boldsymbol{a}) - m(\boldsymbol{a})] \neq 0$ for some $|\boldsymbol{\beta}| \leq n$ such that $\boldsymbol{\beta}(i) > 0$. Therefore

$$\frac{1}{\sigma^2} \sum_{|\boldsymbol{\alpha}| \leq n} \frac{(\mathrm{D}^{\boldsymbol{\alpha}}[f(\boldsymbol{a}) - m(\boldsymbol{a})])^2}{c_{\boldsymbol{\alpha}} \boldsymbol{\lambda}^{\boldsymbol{\alpha}}} + \sum_{|\boldsymbol{\alpha}| \leq n} \log \boldsymbol{\lambda}^{\boldsymbol{\alpha}} \geq \frac{1}{\sigma^2} \frac{(\mathrm{D}^{\boldsymbol{\beta}}[f(\boldsymbol{a}) - m(\boldsymbol{a})])^2}{c_{\boldsymbol{\beta}} \boldsymbol{\lambda}^{\boldsymbol{\beta}}} + \sum_{|\boldsymbol{\alpha}| \leq n} \log \boldsymbol{\lambda}^{\boldsymbol{\alpha}}$$

$$\geq \frac{C}{\lambda_i} + \sum_{|\boldsymbol{\alpha}| \leq n} \log \boldsymbol{\lambda}^{\boldsymbol{\alpha}}$$

for a certain positive constant $C$ and $\lambda_i \leq 1$. Since $1/x + a \log x \to \infty$ for any $a > 0$ as $x \to 0$ from the right, we conclude from the above lower bound that $\tilde{\ell}(\boldsymbol{\lambda}) \to \infty$ as $\lambda_i \to 0$. This concludes the proof. $\qquad\square$

Lemma 3.1 states that $\tilde{\ell}(\boldsymbol{\lambda})$ attains a minimum at $\lambda_i = 0$ when the data are consistent with $m$ being equal to $f$ up to constant along dimension $i$. From Lemma 3.1 and the fact that $\tilde{\ell}(\boldsymbol{\lambda})$ can tend to negative infinity only if a component of $\boldsymbol{\lambda}$ tends to zero we obtain the following theorem, which is essentially a special case of Theorem 5.2 in Karvonen and Oates (2023). See also Proposition 4.3 in Ben Salem et al. (2019).

**Theorem 3.2.** *Suppose that $n \geq 1$ and let $1 \leq i \leq d$. Fix $\lambda_j$ for $j \neq i$. Then*

$$\lambda_{i,\mathsf{ML}} = \arg\min_{\lambda_i \geq 0} \tilde{\ell}((\lambda_1, \ldots, \lambda_d)) = 0$$

*if and only if $\mathrm{D}^{\boldsymbol{\alpha}}[f(\boldsymbol{a}) - m(\boldsymbol{a})] = 0$ for every $|\boldsymbol{\alpha}| \leq n$ such that $\boldsymbol{\alpha}(i) > 0$. In particular, if $d = 1$, then*

$$\lambda_{\mathsf{ML}} = \arg\min_{\lambda \geq 0} \tilde{\ell}(\lambda) = 0$$

*if and only if $f^{(p)}(a) - m^{(p)}(a) = 0$ for every $1 \leq p \leq n$.*

For simplicity, let $d = 1$. If $\lambda = 0$, we see from (2.8) that $P_{n,a}(x, y) = 0$ for all $x, y \in \mathbb{R}$. Moreover, if $f^{(p)}(a) - m^{(p)}(a) = 0$ for every $1 \leq p \leq n$, then

$$s_{n,a}(x) = \sum_{p=0}^{n} \frac{f^{(p)}(a) - m^{(p)}(a)}{p!} (x - a)^p = f(a) - m(a)$$

for every $x \in \mathbb{R}$. That is, $s_{n,a}$ is a constant function. The interpretation of Theorem 3.2 is thus that when the data look like they could have been generated by the function $f(x) = m(x) + c$ for some $c \in \mathbb{R}$ (i.e., by a constant shift of the prior), maximum likelihood estimation returns $\lambda_{\mathsf{ML}} = 0$ because this value of $\lambda$ both explains the data and yield the simplest model, one of zero variance. When the posterior covariance is identically zero, the resulting degenerate posterior $f_{\mathsf{GP}} \mid \boldsymbol{f}_a \sim \mathsf{GP}(f(a) - m(a), 0)$ does not provide useful uncertainty quantification as it is unreasonable to expect perfect predictions from a finite set of data.

### 3.3 On Simultaneous Estimation of $\sigma$ and $\lambda$

The purpose of this section is to demonstrate that simultaneous maximum likelihood estimation of $\sigma$ and $\boldsymbol{\lambda}$ is likely to cause problems. We consider inner product kernels of the form (2.3) with coefficients $c_{\boldsymbol{\alpha}} = c_{|\boldsymbol{\alpha}|}\boldsymbol{\alpha}!/|\boldsymbol{\alpha}|!$ and $n = 1$. Let $\partial_i^p f(\boldsymbol{x})$ denote the $p$th order partial derivative of $f$ at $\boldsymbol{x}$ with respect to the $i$th coordinate.

Note from $c_{\boldsymbol{\alpha}} = c_{|\boldsymbol{\alpha}|}\boldsymbol{\alpha}!/|\boldsymbol{\alpha}|!$ that $c_{\boldsymbol{\alpha}} = c_0$ for $\boldsymbol{\alpha} = \boldsymbol{0}$ and $c_{\boldsymbol{\alpha}} = c_1$ when $|\boldsymbol{\alpha}| = 1$. By differentiating (3.1) with respect to the $i$th component of $\boldsymbol{\lambda}$ we see that to obtain $\boldsymbol{\lambda}_{\mathsf{ML}} = (\lambda_{1,\mathsf{ML}}, \ldots, \lambda_{d,\mathsf{ML}})$ we need to solve

$$\frac{1}{\sigma^2} \sum_{|\boldsymbol{\alpha}| \leq 1} \frac{\boldsymbol{\alpha}(i)(\mathrm{D}^{\boldsymbol{\alpha}}[f(\boldsymbol{a}) - m(\boldsymbol{a})])^2}{c_{\boldsymbol{\alpha}}\boldsymbol{\lambda}^{\boldsymbol{\alpha}}} - \sum_{|\boldsymbol{\alpha}| \leq 1} \boldsymbol{\alpha}(i) = \frac{(\partial_i[f(\boldsymbol{a}) - m(\boldsymbol{a})])^2}{\sigma^2 c_1 \lambda_i} - 1 = 0 \tag{3.3}$$

for each $i = 1, \ldots, d$. Equation (3.3) readily gives the maximum likelihood estimates

$$\lambda_{i,\mathsf{ML}} = \frac{(\partial_i[f(\boldsymbol{a}) - m(\boldsymbol{a})])^2}{\sigma^2 c_1} \tag{3.4}$$

for a fixed $\sigma > 0$. Inserting these to the expression for the maximum likelihood estimate $\sigma_{\mathsf{ML}}^2$ in (3.2) yields

$$\sigma_{\mathsf{ML}}^2 = \frac{1}{d+1}\left(\frac{[f(\boldsymbol{a}) - m(\boldsymbol{a})]^2}{c_0} + \sum_{i=1}^d \frac{(\partial_i[f(\boldsymbol{a}) - m(\boldsymbol{a})])^2}{c_1 \lambda_{i,\mathsf{ML}}}\right) = \frac{1}{d+1}\left(\frac{[f(\boldsymbol{a}) - m(\boldsymbol{a})]^2}{c_0} + d\sigma_{\mathsf{ML}}^2\right),$$

which is solved by $\sigma_{\mathsf{ML}}^2 = [f(\boldsymbol{a}) - m(\boldsymbol{a})]^2/c_0$. By plugging this in (3.4) we obtain the final estimates

$$\sigma_{\mathsf{ML}}^2 = \frac{[f(\boldsymbol{a}) - m(\boldsymbol{a})]^2}{c_0} \quad \text{and} \quad \lambda_{i,\mathsf{ML}} = \frac{c_0}{c_1}\left(\frac{\partial_i[f(\boldsymbol{a}) - m(\boldsymbol{a})]}{f(\boldsymbol{a}) - m(\boldsymbol{a})}\right)^2. \tag{3.5}$$

It is clear what the problem is: If $|f(\boldsymbol{a}) - m(\boldsymbol{a})|$ is small relative to $|\partial_i[f(\boldsymbol{a}) - m(\boldsymbol{a})]|$ for some $i$ (i.e., $m \approx f$ but $\partial_i m \not\approx \partial_i f$ at $\boldsymbol{a}$), the estimate for $\boldsymbol{\lambda}$ becomes large, which may cause numerical problems and yields a large posterior variance. For example, let $d = 1$ and insert the estimates in (3.5) to the posterior variance in (2.8). This gives

$$P_{n,a}(x,x) = \sum_{p=2}^{\infty}\left(\frac{c_0}{[f(a) - m(a)]^2}\right)^{p-1}\left(\frac{[f'(a) - m'(a)]^2}{c_1}\right)^p \frac{c_p}{(p!)^2}(x - a)^{2p}.$$

In practice it is therefore safest to fix one of the parameters and estimate the other. In the examples in Section 6 we fix $\boldsymbol{\lambda}$ and estimate $\sigma$.

## 4 Comparison to the Gaussian Kernel

It is common to condition a Gaussian process defined by the Gaussian kernel on evaluations of a function and its derivative at a number of different points (Solak et al., 2002; Prüher and Särkkä, 2016; Wu et al., 2017). However, no convenient expressions for the posterior mean and variance are available in this setting. The purpose of this section is to exploit an expression from Xu and Stein (2017) and derive explicit expressions for the mean and variance when in the setting of Section 2.2 (i.e., when the data consist of derivative evaluations at a single point). Because the Gaussian kernel is not a Taylor kernel, the posterior mean and variance, although available in closed form, are more complicated than the ones for Taylor kernels in (2.8).

Let $\|\cdot\|_{\boldsymbol{\lambda}} = \langle\cdot, \cdot\rangle_{\boldsymbol{\lambda}}$. It is well known that the ubiquitous Gaussian kernel

$$R(\boldsymbol{x}, \boldsymbol{y}) = \exp\left(-\frac{1}{2}\|\boldsymbol{x} - \boldsymbol{y}\|_{\boldsymbol{\lambda}}^2\right)$$

is closely connected to the exponential kernel in (2.4) via the equation

$$R(\boldsymbol{x}, \boldsymbol{y}) = \exp\left(-\frac{1}{2}\big[\langle\boldsymbol{x}, \boldsymbol{x}\rangle_{\boldsymbol{\lambda}} - 2\langle\boldsymbol{x}, \boldsymbol{y}\rangle_{\boldsymbol{\lambda}} + \langle\boldsymbol{x}, \boldsymbol{x}\rangle_{\boldsymbol{\lambda}}\big]\right) = \exp\left(-\frac{1}{2}\|\boldsymbol{x}\|_{\boldsymbol{\lambda}}^2\right)\exp(\langle\boldsymbol{x}, \boldsymbol{y}\rangle_{\boldsymbol{\lambda}})\exp\left(-\frac{1}{2}\|\boldsymbol{y}\|_{\boldsymbol{\lambda}}^2\right).$$

Given such a relationship to a Taylor kernel it should come as no surprise that, given derivative data, the posterior mean and covariance in (2.6) are available in closed form for the Gaussian kernel—even though the matrix $\boldsymbol{R_a}$ is not diagonal.

Let us consider the univariate case. We get

$$\mathrm{D}_y^i \mathrm{D}_x^j R(x,y)\big|_{\substack{x=a\\y=a}} = (-1)^i \mathrm{D}_z^{i+j} \exp\left(-\frac{\lambda^2}{2}z^2\right)\bigg|_{z=0} = (-1)^i \sum_{p=0}^{\infty}(-1)^p \frac{\lambda^{2p}}{2^p p!}\mathrm{D}_z^{i+j}z^{2p}\big|_{z=0}.$$

When $i+j$ is odd, all derivatives in the sum vanish, so that in this case $\mathrm{D}_y^i \mathrm{D}_x^j R(x,y)\big|_{\substack{x=a\\y=a}} = 0$. If $i+j = 2k$ for $k \in \mathbb{N}_0$, we have

$$\mathrm{D}_y^i \mathrm{D}_x^j R(x,y)\big|_{\substack{x=a\\y=a}} = (-1)^{i+k}\frac{\lambda^{2k}(2k)!}{2^k k!}.$$

This provides us with a relatively straightforward expression for the matrix $\boldsymbol{R_a}$ in (2.7). From the Rodrigues' formula $\mathrm{H}_p(x) = (-1)^p e^{x^2/2}\mathrm{D}_x^n e^{-x^2/2}$ for the probabilist's Hermite polynomials it is easy to compute $\boldsymbol{r_a}(x)$:

$$(\boldsymbol{r_a}(x))_i = \mathrm{D}_y^i R(x,y)\big|_{y=a} = \mathrm{D}_y^i \exp\left(-\frac{\lambda^2}{2}(x-y)^2\right)\bigg|_{y=a} = \lambda^i \exp\left(-\frac{\lambda^2}{2}(x-a)^2\right)\mathrm{H}_i\big(\lambda(x-a)\big). \quad (4.1)$$

Finding $s_{n,a}$ and $P_{n,a}$ requires the inverse of $\boldsymbol{R_a}$. Fortunately, Xu and Stein (2017, Proposition 3.2) have computed the Cholesky decomposition of the inverse of $\boldsymbol{R_a}$.[1] The inverse of $\boldsymbol{R_a}$ has the Cholesky decomposition $\boldsymbol{R_a^{-1}} = \boldsymbol{LL}^{\mathsf{T}}$, where $\boldsymbol{L} \in \mathbb{R}^{(n+1)\times(n+1)}$ is a lower triangular matrix with non-zero elements $(\boldsymbol{L})_{ij} = \sqrt{i!}/(\lambda^j j!(i-j)!!)$ when $i \geq j$ and $i+j$ is even. Here $i!!$ is the double factorial, the product of positive integers up to $i$ that have the same parity as $i$. Thus

$$(\boldsymbol{R_a^{-1}})_{ij} = (\boldsymbol{LL}^{\mathsf{T}})_{ij} = \lambda^{-(i+j)}Q(i,j), \quad \text{where} \quad Q(i,j) = \sum_{\substack{p=\max\{i,j\}\\p+i \text{ is even}}}^{n}\frac{p!}{i!j!(p-i)!!(p-j)!!}. \quad (4.2)$$

Consequently, inserting (4.1) and (4.2) in (2.6) gives the convenient closed form expressions

$$s_{n,a}(x) = \sum_{i=0}^{n}\sum_{j=0}^{n}(\boldsymbol{r_a}(x))_i(\boldsymbol{R_a^{-1}})_{ij}f^{(j)}(a) = \exp\left(-\frac{\lambda^2}{2}(x-a)^2\right)\sum_{j=0}^{n}\frac{f^{(j)}(a)}{\lambda^j j!}\sum_{i=0}^{n}\frac{Q(i,j)}{i!}\mathrm{H}_i\big(\lambda(x-a)\big)$$

and

$$P_{n,a}(x,y) = \exp\left(-\frac{\lambda^2}{2}(x-y)^2\right)\left(1 - e^{-\lambda^2(x-a)(y-a)}\sum_{i=0}^{n}\sum_{j=0}^{n}\frac{Q(i,j)}{i!j!}\mathrm{H}_i\big(\lambda(x-a)\big)\mathrm{H}_j\big(\lambda(y-a)\big)\right).$$

In particular, the posterior variance is

$$P_{n,a}(x,x) = 1 - e^{-\lambda^2(x-a)^2}\sum_{i=0}^{n}\sum_{j=0}^{n}\frac{Q(i,j)}{i!j!}\mathrm{H}_i\big(\lambda(x-a)\big)\mathrm{H}_j\big(\lambda(x-a)\big).$$

These expressions resemble those in (2.8) for Taylor kernels. However, a notable difference is that for Taylor kernels the posterior variance blows up as $|x-a|$ grows but for the Gaussian kernel the variance tends to a constant as $|x-a| \to \infty$. As discussed in Section 2.3, both posterior variances which blow up and those that remain bounded have their uses. Similarly, the posterior mean for Taylor kernels is unbounded, while for the Gaussian kernel the mean reverts to zero (i.e., the prior mean; recall that we set $m \equiv 0$) far away from $a$.

## 5 General Orthogonal Data

In this section we discuss how simple posterior formulae analogous to those derived in Section 2.3 are available for any data that are orthogonal in the sense that the data are obtained by taking RKHS inner products of $f$ with respect to functions that are orthogonal in the RKHS.

---

[1]Note that the denominator in Equation (3.1) of Xu and Stein (2017) should have $(i-j)!!$ in the place of $(i-j)$.

### 5.1 Generic Construction

Let $\Omega$ be an arbitrary non-empty set, $\mathcal{P}$ a countable index set, and $(\phi_p)_{p \in \mathcal{P}}$ a collection of linearly independent basis functions on $\Omega$ such that $\sum_{p \in \mathcal{P}} |\phi_p(x)|^2 < \infty$ for every $x \in \Omega$. We may then define (at least formally) a Gaussian process $f_{\mathsf{GP}}$ on $\Omega$ by setting $f_{\mathsf{GP}}(x) = \sum_{p \in \mathcal{P}} Z_p \phi_p(x)$ for every $x \in \Omega$, where $Z_p$ are i.i.d standard normal random variables. It is then straightforward to compute that

$$\mathbb{E}[f_{\mathsf{GP}}(x)] = 0 \quad \text{and} \quad R(x,y) = \text{Cov}[f_{\mathsf{GP}}(x), f_{\mathsf{GP}}(y)] = \sum_{p \in \mathcal{P}} \phi_p(x)\phi_p(y). \tag{5.1}$$

The kernel $R$ is positive-semidefinite. Assume that $(\phi_p)_{p \in \mathcal{P}}$ are an orthonormal basis of the RKHS $\mathcal{H}(R)$ (see Section 2.4 for RKHSs).[2] Then each $f \in \mathcal{H}(R)$ has the pointwise convergent expansion $f(x) = \sum_{p \in \mathcal{P}} f_p \phi_p(x)$ for the coefficients $f_p = \langle f, \phi_p \rangle_{\mathcal{H}(R)}$. Suppose that one observes $\gamma_p f_p$ for $p$ in a finite collection $\mathcal{N} \subset \mathcal{P}$ of indices and some constants $\gamma_p$. These data are *orthogonal* because they are obtained by taking inner products of $f$ with a collection of functions $\gamma_p \phi_p$ that are pairwise orthogonal in the RKHS. That is, each observation $\gamma_p f_p$ may be written as

$$\gamma_p f_p = \langle f, \gamma_p \phi_p \rangle_{\mathcal{H}(R)} \quad \text{for functions such that} \quad \langle \gamma_p \phi_p, \gamma_q \phi_q \rangle_{\mathcal{H}(R)} = 0 \ \text{ if } \ p \neq q.$$

The orthogonality of $\gamma_p \phi_p$ implies that the corresponding covariance matrix is diagonal. A derivation similar to that in Section 2.3 then shows that the posterior mean and covariance are simply

$$s(x) = \sum_{p \in \mathcal{N}} f_p \phi_p(x) = \sum_{p \in \mathcal{N}} \gamma_p f_p \frac{\phi_p(x)}{\gamma_p} \quad \text{and} \quad P(x,y) = \sum_{p \in \mathcal{P} \setminus \mathcal{N}} \phi_p(x)\phi_p(y) \tag{5.2}$$

for all $x, y \in \Omega$. See (Wendland, 2005, Ch. 16) and (Oettershagen, 2017, Cor. 3.6) for general formulae when the data consists of applications to $f$ of arbitrary linear functionals. Orthogonal data are known to be optimal in a certain sense and settings (Novak and Woźniakowski, 2008, Sec. 4.2.3). To connect (5.2) to the derivations for Taylor kernels, suppose for instance that $f \colon \mathbb{R} \to \mathbb{R}$ has the Taylor expansion

$$f(x) = \sum_{p=0}^{\infty} \frac{f^{(p)}(0)}{p!} x^p = \sum_{p=0}^{\infty} f_p \phi_p(x), \quad \text{where} \quad \phi_p(x) = \frac{\sigma\sqrt{c_p \lambda^p}}{p!} x^p \quad \text{and} \quad f_p = \frac{f^{(p)}(0)}{\sigma\sqrt{c_p \lambda^p}}.$$

In this case we therefore have the index set $\mathcal{P} = \mathbb{N}_0$. Observing the $N$ first derivatives $f^{(p)}(0) = \gamma_p f_p$, so that $\mathcal{N} = \{0, 1, \dots, N\}$ and $\gamma_p = \sigma\sqrt{c_p \lambda^p}$, and using (5.2) yields (2.8) with $d = 1$, $a = 0$, and $m \equiv 0$. Moreover, $\text{Cov}[f_{\mathsf{GP}}(x), f_{\mathsf{GP}}(y)] = \sum_{p=0}^{\infty} \phi_p(x)\phi_p(y) = K(x,y)$ for $K$ the univariate Taylor kernel in (1.2). We mention two other of examples orthogonal data.

### 5.2 Mehler Kernel

Let $\mathcal{P} = \mathbb{N}_0$ and $\Omega = \mathbb{R}$. Let $\mathrm{H}_p$ be the $p$th probabilist's Hermite polynomial and $\rho \in (0, 1)$. Set $\phi_p(x) = \sigma\sqrt{\rho^p (p!)^{-1}} \mathrm{H}_p(x)$. Then Mehler's formula yields the *Mehler kernel*

$$R(x,y) = \sum_{p=0}^{\infty} \phi_p(x)\phi_p(y) = \sigma^2 \sum_{p=0}^{\infty} \frac{\rho^p}{p!} \mathrm{H}_p(x)\mathrm{H}_p(y) = \frac{\sigma^2}{\sqrt{1-\rho^2}} \exp\left(-\frac{\rho^2(x^2+y^2) - 2\rho xy}{2(1-\rho^2)}\right).$$

See Irrgeher and Leobacher (2015) and Oettershagen (2017, Sec. 3.6.4) for the Mehler kernel in the context of kernel-based approximation. If $f(x) = \sum_{p=0}^{\infty} f_p \phi_p(x)$, then by the orthogonality with respect to Gaussian integration, other basic properties of the Hermite polynomials, and properties of Gaussian integrals of derivatives (e.g., Bogachev, 1998, Rmk. 1.3.5),

$$\gamma_p f_p = \frac{1}{\sqrt{2\pi}} \int_{\mathbb{R}} f(x) \mathrm{H}_p(x) \exp\left(-\frac{x^2}{2}\right) \mathrm{d}x = \frac{1}{\sqrt{2\pi}} \int_{\mathbb{R}} f^{(p)}(x) \exp\left(-\frac{x^2}{2}\right) \mathrm{d}x, \quad \text{where} \quad \gamma_p = \sigma \rho^p.$$

Here orthogonal data are therefore obtained by Gaussian integration of the derivatives of $f$.

---

[2] Any functions $(\phi_p)_{p \in \mathcal{P}}$ for which the expansion of the kernel $R$ in (5.1) converges pointwise are a *Parseval frame* for the RKHS $\mathcal{H}(R)$ (Paulsen and Raghupathi, 2016, Thm. 2.10).

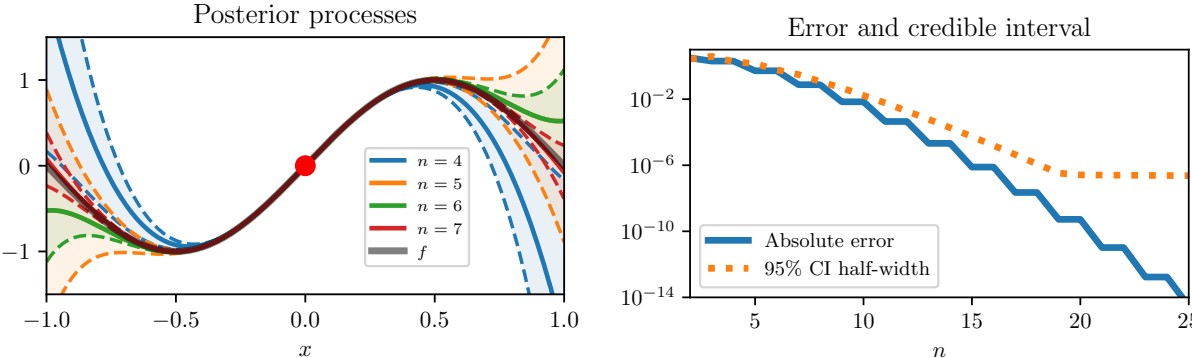

Figure 2: *Left:* Posterior means and 95% credible intervals given derivative data for $f(x) = \sin(\pi x)$ at $a = 0$. The zero-mean prior uses the exponential kernel $K(x,y) = \sigma^2 \exp(\lambda xy)$ with $\lambda = 3/2$ and scale $\sigma$ set using maximum likelihood. *Right:* Maximal absolute errors $\max_{x \in [-1,1]} |f(x) - s_{n,a}(x)|$ and half-widths $1.96 \times \max_{x \in [-1,1]} \sqrt{P_{n,a}(x,x)}$ of the 95% credible interval over the domain $\Omega = [-1,1]$.

### 5.3 Periodic Kernel

Let $\mathcal{P} = \mathbb{Z}$, $\Omega = [0,1]$, and $s \in \mathbb{N}$. Set $\phi_p(x) = \sigma\sqrt{2}(2\pi p)^{-s}\cos(2\pi px)$ and $\phi_{-p}(x) = \sigma\sqrt{2}(2\pi p)^{-s}\sin(2\pi px)$ for $p \in \mathbb{N}$. Moreover, set $\phi_0 \equiv \sigma$. Then we obtain the *periodic Sobolev kernel* (or the *Korobov kernel*)

$$R(x,y) = \sum_{p \in \mathbb{Z}} \phi_p(x)\phi_p(y) = \sigma^2\left(1 + 2\sum_{p=1}^{\infty} \frac{1}{(2\pi p)^{2s}}\cos(2\pi p(x-y))\right) = \sigma^2\left(1 + \frac{(-1)^{s+1}}{(2s)!}\mathrm{B}_{2s}(|x-y|)\right) \quad (5.3)$$

for $x, y \in [0,1]$, where $\mathrm{B}_{2s}$ is the Bernoulli polynomial of degree $2s$; see, for example, Wahba (1990, Sec. 2.1). If $f\colon [0,1] \to \mathbb{R}$ has the expansion $f(x) = \sum_{p \in \mathbb{Z}} f_p \phi_p(x)$, then it is straightforward to compute that

$$\gamma_p f_p = 2\int_0^1 f(x)\cos(2\pi px)\,\mathrm{d}x \quad \text{and} \quad f_{-p} = 2\int_0^1 f(x)\sin(2\pi px)\,\mathrm{d}x \quad \text{for} \quad p \in \mathbb{N} \tag{5.4}$$

for $\gamma_p = \sigma\sqrt{2}(2\pi p)^{-s}$ and $\gamma_0 f_0 = \int_0^1 f(x)\,\mathrm{d}x$ for $\gamma = \sigma$. These orthogonal data are the Fourier coefficients.

## 6 Two Toy Examples

This section contains two numerical toy examples. Figure 2 displays a number of posterior processes and the behaviour of maximal error and standard deviation when a zero-mean Gaussian process with the Taylor kernel $K(x,y) = \sigma^2 \exp(\lambda xy)$ with $\lambda = 3/2$ is used to infer the function $f(x) = \sin(\pi x)$ based on noiseless derivative evaluations at $a = 0$, as described in Section 2. See also Figure 1. The scaling parameter $\sigma$ was taken to be the maximum likelihood estimate in (3.2). From the right panel we see that the Gaussian process model is well-calibrated in the weak sense that, except for small $n$, $f(x)$ is never further away from the posterior mean than maximal half-width of the 95% credible interval over the domain $\Omega = [-1,1]$ of interest: $\max_{x \in [-1,1]} |f(x) - s_{n,a}(x)| \leq 1.96 \times \max_{x \in [-1,1]} \sqrt{P_{n,a}(x,x)}$.

Our second example uses the periodic kernel (5.3) with $s = 2$ and scaled Fourier data in (5.4), so that the posterior mean and covariance are given by (5.2). We use index sets of the form $\mathcal{N} = \{-n, \ldots, -1, 0, 1, \ldots, n\}$ for $n \in \mathbb{N}$ and again use maximum likelihood to set the scaling parameter, which in this case simply yields $\sigma_{\mathsf{ML}}^2 = \frac{1}{2n+1}\sum_{p=-n}^{n}(\gamma_p f_p)^2$. The function being inferred is $f(x) = \exp(x)$, and we compute that

$$\gamma_p f_p = s_p[2e\pi p\sin(2\pi p) + e\cos(2\pi p) - 1] \quad \text{and} \quad \gamma_{-p} f_{-p} = s_p[2\pi p + e\sin(2\pi p) - 2e\pi p\cos(2\pi p)]$$

for $p \in \mathbb{N}$, where $s_p = 2/(4\pi^2 p^2 + 1)$, and $\gamma_0 f_0 = e - 1$. Figure 3 depicts some of the resulting posterior processes. Except at the boundaries where the Gibbs phenomenon caused by the non-periodicity of $f$ occurs, the posteriors fare well and appear to provide reasonable quantification of predictive uncertainty.

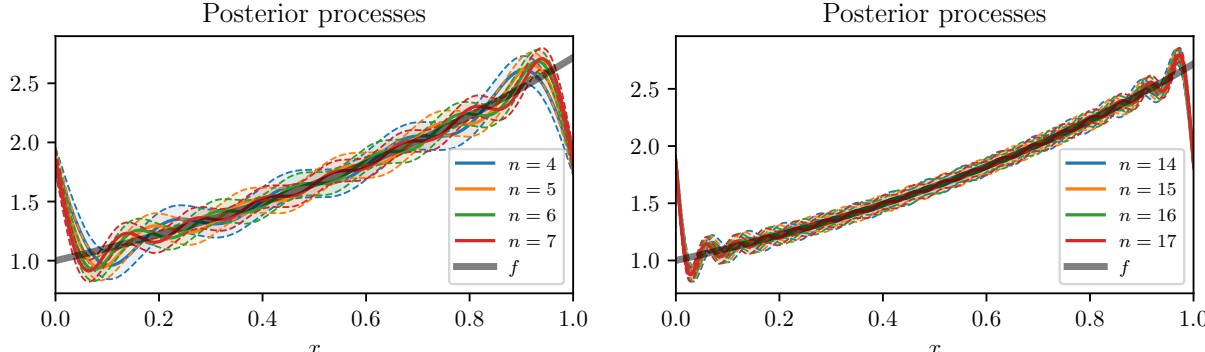

Figure 3: Posterior means and 95% credible intervals given Fourier data for $f(x) = \exp(x)$ on $\Omega = [0, 1]$. The zero-mean prior uses the periodic kernel in (5.3) with $s = 2$ and scale $\sigma$ set using maximum likelihood.

## 7   Conclusion

We have proposed a Gaussian process model based on Taylor kernels which gives rise to a probabilistic version of the classical Taylor expansion when the data consist of derivative evaluations. Using Taylor kernels in Bayesian optimisation (Snoek et al., 2012) would be an interesting future application, where they might be expected to inherit properties from both standard Bayesian optimisation algorithms based on commonly used stationary kernels, such as the Gaussian and Matérns, and classical optimisation algorithms. Because their uncertainty explodes away from the expansion point, Taylor kernels might prove a useful alternative to stationary kernels which have a tendency to be over-exploitative in Bayesian optimisation (Bull, 2011).

### Acknowledgments

TK was supported by the Academy of Finland postdoctoral researcher grant #338567 "Scalable, adaptive and reliable probabilistic integration". FT was partially supported by the Wallenberg AI, Autonomous Systems and Software Program (WASP) funded by the Knut and Alice Wallenberg Foundation. We thank Chris J. Oates, Onur Teymur, and Matt Graham for helpful comments on an early version of this article.

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
