# OpenReview forum: "A probabilistic Taylor expansion with Gaussian processes"
_TMLR — Accepted by TMLR_

### Review · Reviewer_Hfrv · 2023-06-22

**Summary Of Contributions:**

The paper discusses the connection of Gaussian process covariance functions and Taylor series. It prooves certain connections between convergence radii, discusses fitting hyperparameter, in particular those corresponding to Taylor coefficients, and it proves and generalizes orthogonality conditions.

**Audience:**

Yes

**Claims And Evidence:**

Yes

**Requested Changes:**

 - The characterization of GPs at the beginning of Subsection 2.2 is wrong. It is only correct, if you also demans the subsequent properties about evaluations of GPs.
 - Also in section 2.2, you probably want more than "R is n times differentiable in both arguments", but also that all mixed derivatives exist (and perhaps even continuity of those).
 - As described above, the weakness about non-observable hyperparameters and error bounds could be stressed more.

**Strengths And Weaknesses:**

Strengths:
 - The paper has well-chosen graphical examples.
 - While not being mathematical complicated, the results are mathematically interesting and well-interpreted.
 - The paper is clear in its language. Precice, when it needs to be; while still giving intution whenever possible.
 - The paper checks the boxes in correctness and suitable citations.
 - All claims are verified, mostly theoretically, but also with (few, due to space constraints) examples.
 - The paper was interesting to read to me and I am certain that it is interesting to be read in a wider machine learning audience.

Sorry for the boring review. In my opinion, the paper is mostly ready to be published, apart from a few minor suggestions below.

Weaknesses:
 - One weakness for practical application is the difficulty when applying the dicussed covariance functions in practice. The hyperparameters corresponding to higher order Taylor coeffcients are not determinable. Since the error estimates depend strongly on these non-observable values, the error estimates are mostly guesses - even though they work out nicely in the examples. While not hidden, this point could be stressed or more discussed in the paper.

Questions:
 - Why is $c_\alpha>0$ infinitely often? Why exclude polynomial kernels from your class?
 - Why do you need the hyperparameter lambda? Why not subsume it into the $c_\alpha$'s?

---

> ### Author Response · Authors · 2023-07-21
> **Response to Hfrv**
>
> Thank you for the comments.
>
> - Why is $c_\alpha > 0$ infinitely often? Why exclude polynomial kernels from your class?
>
> We do this because polynomial kernels are merely positive-semidefinite, which means that in the noiseless setting these kernels can be used only when the number of data points does not exceed the number positive coefficients. Excluding this case simplifies presentation. We have added a remark about this at the end of Section 2.3, as well as a short sentence in Section 2.1.
>
> While doing this we also noticed that an assumption about the positivity of the coefficients was missing. We have added this in Section 2.1 (see also the new remark in Section 2.3).
>
> - Why do you need the hyperparameter lambda? Why not subsume it into the $c_\alpha$'s?
>
> This parameter plays a role that is somewhat analogous to that of the lengthscale parameter in stationary kernels. For this reason alone we feel that its inclusion is justified. Furthermore, Section 3.2 contains a result on estimation of $\lambda$ that we find very interesting. We have added some commentary on the parameters and their relation to $c_\alpha$'s in Section 2.1.
>
> - The characterization of GPs at the beginning of Subsection 2.2 is wrong. It is only correct, if you also demans the subsequent properties about evaluations of GPs.
>
> We have modified the presentation at the beginning of Section 2.2.
>
> - Also in section 2.2, you probably want more than "R is n times differentiable in both arguments", but also that all mixed derivatives exist (and perhaps even continuity of those).
>
> We have included what we mean by the kernel being $n$ times differentiable in both arguments. We do not think that the continuity of the highest order derivatives is necessary here.
>
> - As described above, the weakness about non-observable hyperparameters and error bounds could be stressed more.
>
> We have included some brief commentary about this in Section 2.4.

---

> ### Comment · Reviewer_Hfrv · 2023-07-21
> **Response to the responses**
>
> I thank the authors regarding their clear answer and changes.
>
> I am positive that my comments are adequately adressed. And I agree with most changes asked by the other reviewers.
>
> What leaves me puzzled is the new section 7, which was added to the reviesed version of the paper.
> - I have the feeling that this section is smuggeled into the paper.
> - I am no expert in optimization algorithms.
> - The suggested optimization algorithm is in no way evaluated.
>
> I am very much interested in the opinion of the other reviewers. I currently think that the paper does not benefit from this section, and instead I advise the remove this section from this paper and work it into a new paper with thorough investigation of the proposed optimization algorithm.
>
> Otherwise, the paper seems ready to be published.

---

> > ### Author Response · Authors · 2023-07-21
> > **Response to Hfrv**
> >
> > Section 7 is smuggled in the sense that it was part of the paper in an earlier submission to a different venue. This section was not initially included in the present submission because we preferred to shorten the paper and keep it more focused.
> >
> > We would be very happy to remove Section 7 if a consensus on this emerges among the reviewers. However, we also feel that Section 7, while not improving the paper quite as much as its length would suggest, does not subtract from the quality of the paper.

---

> > > ### Comment · Reviewer_cMBi · 2023-07-23
> > > **Section 7**
> > >
> > > I'm personally fine with the authors keeping section 7: it's nice to have a concrete, fleshed out example of utility.

---

### Review · Reviewer_zF4Z · 2023-06-26

**Summary Of Contributions:**

- summary

The article establishes a theoretical link between Gaussian Process regression with derivative data and a probabilistic version of Taylor's theorem expansion. The Gaussian Process framework helps quantify the uncertainty of Taylor approximations when the target and reference points are not so close.

**Audience:**

Yes

**Broader Impact Concerns:**

I do not have any ethical concert about this paper.

**Claims And Evidence:**

Yes

**Requested Changes:**

questions:

- What is a possible application of the proposed method?
- How expansive is the computation of the differential data? What is the overall complexity of the proposed approach? Are Taylor kernels computationally cheaper than Gaussian kernels or standard derivative-based regression methods?
- Is assuming the independence of the errors as in Section 2.5 standard? How can this happen, intuitively, when all derivatives are obtained from the same process?
- Which parameters can be estimated (through ML)? Which are fixed by the Taylor structure? Is the "estimation of c_p" equivalent to choosing a specific kernel class? Does c_p need to be an explicit function of p?
- What is the main difference between the proposed approach and existing methods for regression with derivative data?
- Why is the possible uncontrolled behaviour of Taylor Kernels when |x-a| grows a good feature?



**Strengths And Weaknesses:**

- strengths

Developing methods to estimate Taylor's approximation error empirically is a good idea. Interestingly, the uncertainty depends explicitly on the kernel hyperparameters.

Could the provided uncertainty quantification be exploited as a general tool for formal proofs involving the Taylor expansion?

- weaknesses

The clarity of the exposition should be improved. Sometimes, it is hard to understand the goal of an entire section.  For example, introducing RKHS does not seem to be strictly necessary.

The authors should give a practical example where estimating the uncertainty of the Taylor expansion is useful.

It is unclear what is the net contribution of the paper.  Gaussian Process regression with derivative data already exists. But, according to the authors, previous work only focuses on the Gaussian Kernel. Is the novelty here the extension to the more Taylor Kernels defined in Equation 2.3? If so, the authors should point out the advantages of such an extension.

Compared to Gaussian kernels, Taylor kernels may diverge or grow exponentially when |x-a| is large. The authors do not discuss the practical and theoretical implications of this. It would be nice to include a set of conditions (e.g. on the coefficients of the Taylor expansion) such that the good behaviour of Gaussian kernels is reproduced.

The experiments are limited. The very brief experiment section suggests general confusion about the practical impact of the proposed scheme.

---

> ### Author Response · Authors · 2023-07-21
> **Response to zF4Z - part 1**
>
> Thank you for the comments.
>
> - Could the provided uncertainty quantification be exploited as a general tool for formal proofs involving the Taylor expansion?
>
> We find this unlikely. Typically (both here and elsewhere when working with GP uncertainty quantification) one connects uncertainty quantification to some object from approximation theory and exploits the properties of that object rather than the other way around.
>
> - For example, introducing RKHS does not seem to be strictly necessary.
>
> It is true that we could survive without RKHSs - this is in fact an approach that we experimented with but, in the end, found it more convenient to use RKHSs (while strictly confining them to Sections 2.4 and 5). Our use of RKHSs does make these two sections more forbidding to readers less familiar with RKHSs, which is a drawback that we are willing to accept.
>
> - The authors should give a practical example where estimating the uncertainty of the Taylor expansion is useful.
> - The experiments are limited. The very brief experiment section suggests general confusion about the practical impact of the proposed scheme.
> - What is a possible application of the proposed method?
>
> We have added a better motivating example in Section 7. See the response addressed to all reviewers.

---

> ### Author Response · Authors · 2023-07-21
> **Response to zF4Z - part 2**
>
> - It is unclear what is the net contribution of the paper. Gaussian Process regression with derivative data already exists. But, according to the authors, previous work only focuses on the Gaussian Kernel. Is the novelty here the extension to the more Taylor Kernels defined in Equation 2.3? If so, the authors should point out the advantages of such an extension.
> - What is the main difference between the proposed approach and existing methods for regression with derivative data?
>
> The point is that previous work on GP regression with derivative data does not lead to a Taylor expansion. This is for two reasons: (1) the kernels used (typically the Gaussian) are such that this cannot happend and (2) the data consist of a mixture of function and derivative evaluations at different points, whereas here all derivatives are evaluated at the some point. We have tried to clarify this by expanding the last paragraph of Section 1.1 (we have also included some related material at the beginning of Section 4). As we describe in the last paragraph of Section 1.0, the main advantage (as we see it) is that we obtain a probabilistic GP interpretation of the Taylor expansion. Such an interpretation is partially a curiosity, but also fits well within a collection of analogous interpretations of classical numerical methods in the field of probabilistic numerics.
>
> - Compared to Gaussian kernels, Taylor kernels may diverge or grow exponentially when |x-a| is large. The authors do not discuss the practical and theoretical implications of this. It would be nice to include a set of conditions (e.g. on the coefficients of the Taylor expansion) such that the good behaviour of Gaussian kernels is reproduced.
> - Why is the possible uncontrolled behaviour of Taylor Kernels when |x-a| grows a good feature?
>
> We are not sure there is that much to discuss here, at least when it comes to theory. Taylor kernels and stationary kernels such as the Gaussian simply encode different kind of prior information about the function. We have added some commentary to this effect at the end of Section 2.3. In particular, we do not think that the properties of the Gaussian kernel are necessarily good or something that one should strive to reproduce if there is no additional prior information that the function behaves in a stationary manner. As we mention in the conclusion, the explosion of uncertainty may be a good feature if one wants to encourage exploration over exploitation.
>
> The only condition that would guarantee that the posterior mean and variance of a Taylor kernel do not blow is that there are only as many non-zero coefficients $c_\alpha$ as there are elements in the dataset. But that is not interesting because then the posterior variance would be identically zero.
>
> - How expansive is the computation of the differential data? What is the overall complexity of the proposed approach? Are Taylor kernels computationally cheaper than Gaussian kernels or standard derivative-based regression methods?
>
> How expensive derivative data is entirely dependent on the specific application and there is little that we can say about it. We consider computational complexity and comparison to existing kernels and methods relatively unimportant questions because, rather proposing a general-purpose method that is supposed to compete with what has been done before, we are interested in understanding how Gaussian processes can be used to replicate classical Taylor expansions.
>
> Nevertheless, in the specific case of derivative data Theorem 2.1 shows that inference with Taylor kernels is significantly faster than with some "generic" kernel. We now say this after Theorem 2.1.
>
> - Is assuming the independence of the errors as in Section 2.5 standard? How can this happen, intuitively, when all derivatives are obtained from the same process?
>
> This assumption is used mostly out of convenience because no closed form expressions for the posterior mean and variance are available (at least in general) if the noises are assumed correlated. We have added some commentary on this.
>
> - Which parameters can be estimated (through ML)? Which are fixed by the Taylor structure? Is the "estimation of c_p" equivalent to choosing a specific kernel class? Does c_p need to be an explicit function of p?
>
> As explained in Section 3, both $\sigma$ and $\lambda$ may be estimated via ML (though estimating both of them simultanously can be hazardous, as explained in Section 3.3). We've added some commentary in Section 2.1 that tries to make this clearer. Unfortunately, we don't understand the last question about $c_p$.

---

> > ### Comment · Reviewer_zF4Z · 2023-07-24
> > **Thank you for your answer and for updating the paper.**
> >
> > I appreciate the authors added a series of intuitive explanations about the difference between their approach and standard GP regression. New Section 7 is interesting but rather long. I think the paper is publishable even without that section, provided the authors clarify that the paper is more about interpreting existing techniques than proposing new algorithms. My remaining (minor) concern is the assumption of uncorrelated noise. If I understand correctly, evaluating the derivatives at the same point distinguishes the proposed approach from standard GP regression. Does this may make the assumption harder to justify? Where is the assumption used (except for Section 2.5)?

---

> > > ### Author Response · Authors · 2023-07-28
> > > **Second response to zF4Z**
> > >
> > > We have decided to remove Section 7 (see the general response for more details). As suggested, we have added a clear statement about the purpose of the paper in Section 1.2.
> > >
> > > The assumption that the noise is uncorrelated is only used in Section 2.5 because that is the only place where we consider noisy observations - everywhere else in the paper we work in the "interpolatory" setting where the observations are assumed noiseless, which is a setting frequently considered in the GP literature and rather realistic if the observations are, for example, outputs from a computer simulation.
> > >
> > > It is probable that the uncorrelatedness assumption is harder to justify when derivatives are being evaluated at a single point. This is the gist of the sentence "While unrealistic in practice, the assumption that the noise terms are independent allows for explicit computation of the posterior mean and variance." that we added in Section 2.5 in the first revision. Section 2.5 is quite tangential to other parts of the paper, so there is not much more that we find necessary to say.

---

### Review · Reviewer_cMBi · 2023-07-09

**Summary Of Contributions:**

The authors consider using GPs to model epistemic uncertainty when approximating a function locally using a Taylor series expansion. They introduce Taylor kernels for both univeriate and multivariate input GPs. Perhaps the key result is that when using these kernels and conditioning on a number of known derivatives at a point, the resulting posterior GP has mean function equal to the usual Taylor expansion. That in itself of course is not useful, but the GP also provides an analytic expression for the posterior variance, which grows to infinity far from the expansion point. Further theory is developed covering error bound, noisy data, hyperparameter estimation and "orthogonal data". Two small numerical examples are presented.


**Audience:**

Yes

**Claims And Evidence:**

Yes

**Requested Changes:**

Ideally I would like to see a more "real" motivating example where the theory can be put to practice.

Minor:
- I believe K in Theorem 2.1 is missing a K_a - term.
- In the proof for thm 2.1 there is no need to repeat the final s and P equations.
- just before section 5.2: examples OF orthogonal data

**Strengths And Weaknesses:**

The paper is reasonably well written apart from missing a few definitions, e.g. I'm still unclear what "orthogonal data" are. It is on the long side and mathematically dense, I wonder if there are intermediate steps that the authors could move to an appendix to allow the reader to focus on the key results.

I'm unclear if the key result (mentioned above) is novel: if so that is very elegant and in my mind the major strength of the work.

Perhaps the main weakness is that practical utility is not demonstrated well. There are just two toy numerical examples, approximating sin(pi x) and exp(x). Neither test multivariate inputs. Relatedly, their own conclusion is that simultaneous estimation of sigma (signal variance) and lambda (inverse length scale) is problematic. How should this be handled in practice? Would Bayesian inference help (at some computational cost)?

---

> ### Author Response · Authors · 2023-07-21
> **Response to cMBi**
>
> Thank you for the comments.
>
> - The paper is reasonably well written apart from missing a few definitions, e.g. I'm still unclear what "orthogonal data" are.
>
> You are right the the matter of orthogonal data was not properly explained. We have tried to be more explicit in Section 5.
>
> - It is on the long side and mathematically dense, I wonder if there are intermediate steps that the authors could move to an appendix to allow the reader to focus on the key results.
>
> It would be possible to place some of the derivations in an appendix. However, we prefer the current linear structure.
>
> - I'm unclear if the key result (mentioned above) is novel: if so that is very elegant and in my mind the major strength of the work.
>
> We have decided to add some commentary on the novelty at the end of Section 1.0.
>
> - Perhaps the main weakness is that practical utility is not demonstrated well. There are just two toy numerical examples, approximating sin(pi x) and exp(x). Neither test multivariate inputs. Relatedly, their own conclusion is that simultaneous estimation of sigma (signal variance) and lambda (inverse length scale) is problematic. How should this be handled in practice? Would Bayesian inference help (at some computational cost)?
> - Ideally I would like to see a more "real" motivating example where the theory can be put to practice.
>
> In practice we simply suggest fixing one of these parameters and estimating the other. In the examples of Section 6 and the new example of Section 7 we fix $\lambda$ and estimate $\sigma$, which we now mention at the end of Section 3.3.
>
> We do not know if Bayesian inference would help in simultaneous estimation of these parameters. It is possible that a phenomenon similar to how both the scale and lenghtscale cannot be consistently estimated in Matérn by and related models is at play here (see Theorem 1 in the 1991 paper "Asymptotic Properties of a Maximum Likelihood Estimator with Data from a Gaussian Process" by Z. Ying for one of the earliest results on this). However, we are not confident enough include any speculation about this.
>
> We have added a better motivating example in Section 7. See the response addressed to all reviewers.
>
> - Minor:
>
> We have fixed these.

---

### Author Response · Authors · 2023-07-21
**Response to the reviewers**

We thank all reviewers for their comments which, we hope, have resulted in improvements in the article (and have substantially increased its length). In the revised version major additions and changes are marked in red (except for Section 7 which is new in its entirety).

The most important issues, raised by both zF4Z and cMBi, was the lack of "real" and multivariate applications. To address this we have added Section 7 in which we construct a probabilistic version of the standard quadratic trust-region method for local optimisation. In Section 7 our goal is not to propose a method that outperforms some existing competitors but rather to show that useful numerical algorithms can emerge the probabilistic Taylor expansion. In the logistic regression example (Section 7.2) the proposed algorithm performs similarly to the standard trust-region method, which effectively shows that one may alternatively derive the heuristics-based standard method from probabilistic reasoning.

The remainder of our response is contained in the comments to each individual reviewer.

---

### Author Response · Authors · 2023-07-28
**Second revision**

We thank the reviewers for further discussion and have revised the paper accordingly.

Because both zF4Z and Hfrv favoured (or did not mind) removing Section 7, we have done so as this is also our preference. All references to Section 7 and trust-region methods have been removed from the paper. A further edit to Section 1.2 suggested by zF4Z is marked in red.

---

> ### Comment · Reviewer_Hfrv · 2023-08-08
> **Official Recommendation**
>
> I recommend the paper to be published in the current form (without the addded section 7).